## Registered report 

psychology/human–computer interaction

video games, aggression, adolescents, registered report

**Author for correspondence:**
Andrew K. Przybylski
e-mail: andy.przybylski@oii.ox.ac.uk

# Violent video game engagement is not associated with adolescents' aggressive behaviour: evidence from a registered report

Andrew K. Przybylski[1,2] and Netta Weinstein[3]

[1]Oxford Internet Institute, University of Oxford, Oxford OX1 3JS, UK
[2]Department of Experimental Psychology, University of Oxford, Oxford, UK
[3]School of Psychology, Cardiff University, Cardiff, UK

(iD) AKP, 0000-0001-5547-2185

In this study, we investigated the extent to which adolescents who spend time playing violent video games exhibit higher levels of aggressive behaviour when compared with those who do not. A large sample of British adolescent participants ($n = 1004$) aged 14 and 15 years and an equal number of their carers were interviewed. Young people provided reports of their recent gaming experiences. Further, the violent contents of these games were coded using official EU and US ratings, and carers provided evaluations of their adolescents' aggressive behaviours in the past month. Following a preregistered analysis plan, multiple regression analyses tested the hypothesis that recent violent game play is linearly and positively related to carer assessments of aggressive behaviour. Results did not support this prediction, nor did they support the idea that the relationship between these factors follows a nonlinear parabolic function. There was no evidence for a critical tipping point relating violent game engagement to aggressive behaviour. Sensitivity and exploratory analyses indicated these null effects extended across multiple operationalizations of violent game engagement and when the focus was on another behavioural outcome, namely, prosocial behaviour. The discussion presents an interpretation of this pattern of effects in terms of both the ongoing scientific and policy debates around violent video games, and emerging standards for robust evidence-based policy concerning young people's technology use.

# 1. Introduction

Nearly all young people in the developed world now play video games [1,2], and this popularity has driven concerns about the possible negative effects of this recreational activity. Studies polling members of the general public [3–5] as well as scientists [6,7] suggest views concerning the effects of gaming on young people vary widely as a function of demographics and personal experiences with games. Some researchers conclude that gaming has social [1] and cognitive [8] benefits, whereas others argue the medium contributes to mass-shooting events [9–11], and for consistent and strong effects on aggressive behaviour, more broadly [12]. Recently, a series of open letters published by scholars have cautioned the public and policy-makers that both the positive and negative effects of time spent gaming, their addictive potential [13], cognitive benefits [14,15] and aggressive effects [16] may have been overstated.

Like individuals, policy and professional organizations have expressed varied positions regarding video game effects. In general, most organizations' initial guidance was framed by the *precautionary principle*—an approach to mitigating societal harm that puts protections in place when there is a plausible risk. Policy-makers guided by this mindset have discretion to take measures in cases where scientific knowledge about something new is lacking. In line with this principle, some organizations like the American Psychological Association [17] err on the side of caution and warn to limit youngsters' time spent playing video games. Such steps are far from universal as other organizations conducting their reviews of the science, such as the Australian [18] and Swedish [19] government reports, and the APA's own Media Psychology and Technology Division [20], have concluded there is no actionable evidence that aggressive behaviour results from youth gaming. As more nuanced empirical understanding of media effects has emerged, other policy positions, once stridently aligned against gaming and screen time, such as the American Academy of Pediatrics, have softened their prescriptions concerning digital media and psychosocial development [21]. These changes have been reflected in the statutory arena: in 2011, the United States Supreme Court [22] judged that there is insufficient evidence that games cause harm to uphold laws restricting the sale of violent games to minors. These changes in law and policy follow closely from a shifting empirical landscape.

There is a good reason to believe that violent video game engagement might be associated with human aggression, though this idea is a controversial one [23]. To date, the main theoretical framework used to study the links between violent game engagement and aggression has been the general aggression model (GAM; [22]). Briefly, the GAM is an appetitive social learning theory that proposes that repeated exposure to violent media increases the accessibility of aggressive thoughts, which in turn increases the probability of aggressive cognitive schema, emotions and behaviour [24]. Some reviews [25] and recent studies [26] informed by the GAM framework report consistent, though modest, support for the idea that violent gaming is linked to human aggression. This interpretation is not uniform; other analyses of the literature conducted by Sherry [27,28] and Ferguson [29] provide evidence the GAM framing, and the idea that games cause aggression more broadly, is incomplete, not evidenced or flawed. Indeed, motivation research indicates many factors key to understanding games are often overlooked by GAM researchers, such as the observation that aggressive individuals gravitate towards violent games [30], and that violent games might foment player aggression in experimental studies not because they prime aggressive cognitive schema, but rather that they frustrate the basic psychological need for competence [31].

One noteworthy attempt to bring a measure of harmony to the existing literature is that by Hilgard *et al.* [32], who re-analysed widely cited metanalytic data, drawing together results derived from GAM research [25] that form the basis of a number of past and existing policies regarding violent game effects [17]. Their analysis detected the presence of publication bias not uncovered in the original reporting of the data. Upon adjusting for publication bias, the observed aggregate effect sizes relating gaming to aggression were smaller than those originally presented. Worryingly, this analysis also suggested that studies originally deemed to be following best practices showed particularly strong evidence of publication bias. That understood, the naive and corrected estimated effect sizes relating violent video game play to self-reported aggression extracted from this meta-analysis across a total of 37 studies ($n = 29\,113$) were relatively consistent and small to medium [33] ($r = 0.21$; 95% CI = 0.20–0.22) in size.

With this in mind, there is reason to think that outstanding methodological challenges might be inflating this metanalytic estimate. First, there is a noticeable degree of flexibility in how violent game play is operationalized in survey studies. For example, in multiple published studies of gaming effects drawn from the *Effects of Digital Gaming on Children and Teenagers in Singapore* project (EDGCTS; for a list see: https://osf.io/3gdt5/), violent video game engagement has been computed differently across

presentations of findings from the dataset. In one case, the researchers [34] measured violent gaming by combining responses to three questions into a single variable, one about non-violent gaming (reverse scored): 'How often do other players help each other in this game?', and two questions that assessed violent gaming 'How often do you shoot or kill other players in this game?' and 'How often do characters try to hurt each other's feelings in this game?'. Working with the same data, researchers [35] later selected four items to reflect game content, of which two were not part of the original analysis. These were: 'How often do you shoot or kill creatures in this game?' and 'How often do you help others in this game?'. In place of computing one violent game content variable, the researchers created two separate variables for their analysis, one reflecting non-violent content and a second reflecting violent game content. Subsequent work by the same group [36], again using the same data, relied on a single violent gaming construct but used four items without identifying which of the available items reflecting game content were included. This flexibility, described as part of the 'garden of forking paths problem' increases the chance of false-positive results and serves to reduce our confidence in the inferences linking gaming to aggression [37,38].

Adding to problem of survey measurement flexibility is the fact that violent gaming effects research relies on self-reported data entirely provided by young people. Said differently, this work depends on children and young people accurately reporting on their video game play, the level of violent content present in this play and their own trait- or state-level aggression (for an exception, see [28]). This is problematic because studies of young people [39], health [40] and gaming [41] may be susceptible to the so-called *mischievous responding*—a phenomenon in which research participants exaggerate their responses by selecting extreme, and sometimes implausible, response options when providing self-report data. Mischievous responding can have the effect of introducing measurement noise that inflates relations that are logically incoherent or absurd to take at face value. Indeed, it is possible that some might respond to surveys in such a way that both their video game play and their experiences of intimate sexual behaviours are exaggerated. Such a pattern could lead researchers to make the spurious claim that playing the 2004 Xbox game *Spider Man 2* is a significant catalyst for adolescent promiscuity [42].

Measurement flexibility also extends to a number of outcome assessments employed in the gaming literature. In experiments evaluating gaming aggression, methods for computing self-reported measures of aggressive emotions alternate between approaches that use all of the available scale items [31,43] and those that use a subset of items thought by some researchers to be 'most sensitive to an experimental manipulation of video game play' [9]. A similar tractability is present in behavioural measurements of aggression. The most widely used laboratory-based method for measuring aggression, the competitive reaction time task (CRTT; [34]), has been used in more than 125 published papers, and surprisingly, task scores have been quantified in more than 155 different ways in this literature [44]. In many cases [45], more than one computational approach is used to operationalize behavioural aggression in the same paper. For both self-reported and behavioural aggression measures, this flexibility affords otherwise well-meaning researchers the ability to select between different operationalizations of predictors and outcomes until they find a combination in line with their pre-existing biases or theories. Because this work is almost entirely exploratory in nature (i.e. not preregistered), it is difficult to know what to make of studies that report positive findings under these conditions of routinized methodological flexibility [46].

A handful of preregistered studies have rigorously tested the links between violent game play and human aggression [47], and do not detect an effect of brief exposure to violent gaming on aggression in the laboratory. Given these findings and the wider importance of conclusions drawn from this work, it is important to use preregistered study methodology to evaluate whether the existing literature may be under- or over-estimating the extent to which violent video game play relates to aggression. With this in mind, the present study examined the fundamental dynamic of concern in this subfield through a purely confirmatory lens [47] following a registered reports protocol [48].

Our aim was to rigorously test the hypothesis that time spent playing violent video games is positively associated with adolescents' everyday behavioural aggression. The study examined the extent to which there are detectable positive linear [32] and parabolic, 'U'-shaped, relationships [2] linking these factors. To this end, we analysed data collected from a large and representative cohort of British young people and their carers. Of interest was the significance, direction and effects sizes observed between video game engagement, operationalized as time spent playing violent video games, and aggressive behaviour, operationalized using carer reports of adolescents' aggression.

**Table 1.** *A priori* estimation of required sample size.

| | |
|---|---|
| Input | |
| effect size $f^2$ | 0.042 |
| $\alpha$ err prob | 0.05 |
| power ($1 - \beta$ err prob) | 0.99 |
| number of predictors | 1 |
| Output | |
| non-centrality parameter $\lambda$ | 18.46 |
| critical $F$ | 3.86 |
| numerator d.f. | 1 |
| denominator d.f. | 441 |
| total sample size | 443 |
| actual power | 0.99 |

# 2. Material and methods

For this survey-based study, we recruited a large and nationally representative sample of British adolescents, and quantified recent video game play using a gaming engagement measure validated in previous large-scale survey research [49]. We operationalized violent game contents using European Union [50] and North American media rating systems [51]. We measured youth aggression and prosocial behaviour with carer responses using a behavioural screening questionnaire [52] that has been widely employed by researchers, educators and clinicians to assess the psychosocial functioning of children and adolescents ranging in age from 4 to 17 years.

## 2.1. Power analysis for confirmatory hypothesis testing

In line with the best existing meta-analytic evidence [32], our aim was to test the hypothesis that there was a statistically and practically significant effect relating violent video game play to aggressive behaviour. The existing literature suggests that the effect linking violent video game play to aggressive behaviour in cross-sectional research ($n = 29\,113$) is approximately $r = 0.21$ (95% CI = 0.20–0.22). Our study aimed to achieve 99% power for the lower bound of this range, $r = 0.20$, thus maximizing our chance of correctly rejecting the null hypothesis if this effect if present. The result of this *a priori* power analysis, presented in table 1, indicated that a sample size of 443 would be required to attain the desired power level (99%) for the lower bound of the plausible effect size of interest [33].

## 2.2. Study sampling plan

In line with the results of our power analysis, a target sample of 1000 adolescents (500 females, 500 males) was set. A total of 1004 adolescents and an equal number of their carers were recruited to complete online self-report questionnaires. A participant profile of adolescents aged 14 and 15 years living in England, Scotland and Wales was built using geographical and demographic factors based on 2011 United Kingdom Census data. Geographical region, household socioeconomic class, participant age and gender were considered for quotas to match the census data and sampling continued until these quotas were attained. The final sample was evenly divided among 14-year-old ($n = 497$) and 15-year-old adolescents ($n = 507$). Further, 540 participants identified as male, 461 as female and 3 as another gender orientation. The sample was predominantly white as 8.1% of participants reported they were from Black and other minority ethnicities. The total combined household income mirrored the general population and ranged from £6500 (1.9%) to £150 000 or more (2.8%). The sample was recruited partnering with the research firm ICM Unlimited drawing on a participant pool previously used to recruit nationally representative samples for health [53,54] and technological research [55,56]. Participant consent was received using a double opt-in procedure that was part of a larger study conducted in early March 2018 to survey the online lives and behaviours of British youth. After

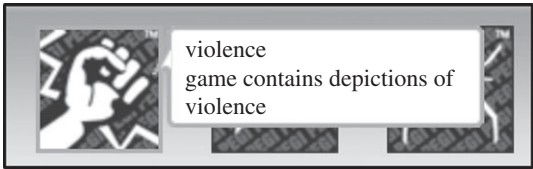

**Figure 1.** Games featuring this PEGI violent content badge were coded as having violent content.

providing consent for their own participation and consent for their children to participate, carers completed demographic questions as well as the criterion variable. After completing their portion of the study, carers were asked to leave the room and adolescents then completed their portion of the study. Adolescent participants registered their own consent and completed personality and gaming behaviour questions.

## 2.3. Research ethics and open science practices

The study underwent ethical review and received approval from the Central University Ethics Committee of the University of Oxford (C1A17023), and all study materials are available for download on the Open Science Framework [57].

## 2.4. Computation of the criterion variable: aggressive behaviour

Carer participants were asked to complete the widely used Strengths and Difficulties Questionnaire (SDQ; [49]) as a measure of adolescents' recent aggressive behaviours. The conduct problems subscale of the SDQ was the study's outcome variable, as it has been extensively used to measure aggressive behaviour and interpersonal aggression across a wide range of cultures in community [58], academic [59] and clinical [60] settings. Carers were asked to provide responses on the basis of their child's aggressive behaviours in the past month. They selected one of three options: '*Not True*' (coded 1), '*Somewhat True*' (coded 2) or '*Certainly True*' (coded 3) to rate five items which reflected conduct problems they had observed in their child, e.g. 'Often fights with other children or bullies them'. In line with past research [61], individual scores were computed by reverse scoring responses to the negatively worded statements and adding this score to the values of the positively worded items (see electronic supplementary material, appendix A). Individuals' scores on the five-item conduct problems subscale of the SDQ ranged from 5 to 15 ($M = 6.93$, s.d. $= 2.20$, $\alpha = 0.76$)

## 2.5. Computation of the explanatory variable: violent video game engagement

Violent video game play scores were computed for each participant using a combination of self-report and objective data. This was done by combining information from five variables (see electronic supplementary material, appendices B and C). Variables 1–4 were used to determine if the games played by the adolescent have violent content and variable 5 was used to determine how much time adolescents devoted to violent play. Participants were asked if they play games (variable 1), for the names of the three games they played most in the past month (variable 2; providing they answered in the affirmative to variable 1), the gaming or computer system used to play these games (variable 3) and whether these games were played in a single or multiplayer mode (variable 4). This information was then used by a coder who was blind to the purpose of the study to find the game's entry on the Pan European Game Information (PEGI) website for confirmatory hypothesis testing. This rating was complemented with data derived from the game's entry on the Entertainment Software Rating Board (ESRB) website, which was used for an exploratory analysis. For example, the entry for Grand Theft Auto V for PC [62] is coded as violent as players see and enact virtual violence during play (figure 1). A total of 1596 games were named and successfully coded for violent content. Of these nearly two in three had a violent content badge and were coded 1 ($k = 1033$), while the remaining games did not have a violent content badge and were coded zero 0 ($k = 560$). In line with recent large-scale research asking this age group about the frequency of gaming activity, adolescents responded to a fifth item, a 9-point scale ranging from '*none at all*' (coded 0) to '*about 7 or more hours a day*' (coded 8) as a frame for reporting the frequency of their play (electronic supplementary material, appendix C; [2]) for each game they identified playing. The amount of time playing each game was multiplied by the PEGI

content coding for the game. Following this plan, we found participants were moderately engaged in violent games, devoting an average of two hours to them on a typical day ($M = 2.07$, s.d. $= 2.30$).

## 2.6. Computation of control variables: trait-level aggression

Individual differences in trait-level aggression were assessed through adolescent self-reports derived from an abbreviated form of the Buss–Perry aggression scale [63]. Adolescents were asked to rate 12 items (electronic supplementary material, appendix D) in terms of how characteristic each is of each using a 5-point scale ranging from '*very unlike me*' (coded 1) to '*very like me*' (coded 5). Four scores were computed for each participant by averaging across responses for the relevant facet; three items for physical aggression ($M = 2.20$, s.d. $= 1.14$, $\alpha = 0.82$), e.g. 'Given enough provocation, I may hit another person', three for verbal aggression ($M = 2.55$, s.d. $= 1.10$, $\alpha = 0.84$), e.g. 'I can't help getting into arguments when people disagree with me,' three for anger ($M = 2.41$, s.d. $= 1.11$, $\alpha = 0.85$), e.g. 'Sometimes I fly off the handle for no good reason,' and three for hostility ($M = 2.60$, s.d. $= 1.08$, $\alpha = 0.85$), e.g. 'I wonder why sometimes I feel so bitter about things'.

## 2.7. Check question: subjective game engagement

Adolescent participants were also asked to rate the extent to which they agreed with the statement: 'I spend a lot of time playing video games' (electronic supplementary material, appendix E) using a 5-point scale that ranged from 1 '*Strongly disagree*' to 5 '*Strongly agree*'. We expected and found that responses to this item were positively correlated with participant estimates of the time they spent playing violent games on a typical day, $r_{768} = 0.47$, $p < 0.001$.

## 2.8. Consideration of outcome-neutral conditions

We judged that a large-scale cross-sectional survey was well-suited to test our hypotheses. Unlike many of the methodologically flexible behavioural (CRTT; [35,36]) or self-reported [9] assessments on which previous work relies, our criterion variable—the Strengths and Difficulties Questionnaire—was completed by carers (not adolescent participants) and has been extensively used to assess interpersonal aggression across a wide range of cultures in community [58], academic [59] and clinical [60] settings, and in more than 4000 studies in 95 countries. Our data supported this decision, as the reliability of the scale was relatively high and in line with previous SDQ work [59]. Further, the explanatory variable, violent video game play, reflected the level of exposure young people have to violent games by combining objective assessments of both time spent playing and violent content (official game ratings) across a broad range of popular games. Most games could be easily categorized by the coder, and this approach contrasts with research practices that either: (i) assign participants to play 'off the shelf' games that broadly represent the categories of violent and non-violent play, leaving open the possibility of other game characteristics such as difficulty having a confounding role [28] or (ii) depend on the accuracy of participants providing subjective ratings of the violence present in their own gaming [36]. Studies also indicate that violent video games are regularly played by both adolescent boys (66–78%) and girls (21–33.6%) [64,65], and observations from our data largely mirrored these statistics. A total of 48.8% of female participants and 68.0% of males played at least one violent game in the past month. Similarly, past research indicates that levels of daily video game play [2], violent game play [66] and aggressive behaviour should all be higher in male adolescents [58]. To examine this issue, we tested relations with gender, and observed higher levels of overall gaming, $t_{764} = 3.71$, $p < 0.001$, violent gaming, $t_{764} = 4.09$, $p < 0.001$ and aggressive behaviour, $t_{999} = 2.98$, $p < 0.001$, in adolescent boys as compared to adolescent girls. Given this, we controlled for variability in gender in our hypothesis testing as planned, although gender differences were not a key question within the current research.

# 3. Results

## 3.1. Preliminary analyses

Table 2 presents the zero-order correlations observed between the variables of interest detailed in the analysis plan. In line with expectations, aggressive behaviour was positively associated with all of the

**Table 2.** Observed zero-order correlations between study variables.

| | 1. | 2. | 3. | 4. | 5. | 6. | 7. | 8. | 9. |
|---|---|---|---|---|---|---|---|---|---|
| 1. female | — | | | | | | | | |
| 2. aggressive behaviour | −0.094** | — | | | | | | | |
| 3. overall game engagement | −0.133** | 0.111** | — | | | | | | |
| 4. plays any violent games | −0.194** | −0.149** | 0.242** | — | | | | | |
| 5. violent game engagement | −0.148** | 0.078* | 0.851** | 0.461** | — | | | | |
| 6. trait physical aggression | −0.111** | 0.622** | 0.135** | −0.063* | 0.108** | — | | | |
| 7. trait verbal aggression | −0.013 | 0.592** | 0.083* | −0.034 | 0.064 | 0.751** | — | | |
| 8. trait anger | −0.016 | 0.616** | 0.099** | −0.056 | 0.086* | 0.769** | 0.854** | — | |
| 9. trait hostility | 0.004 | 0.538** | 0.057 | −0.067* | 0.028 | 0.687** | 0.773** | 0.756** | — |
| 10. subjective game engagement | −0.341** | 0.247** | 0.469** | 0.178** | 0.434** | 0.289** | 0.260** | 0.244** | 0.198** |

** $p < 0.001$ and * $p < 0.01$.

observed variables ($|r|s = 0.08–0.62$), and subjective game engagement was positively correlated with overall levels of game engagement, any violent game engagement and overall levels of violent game engagement ($rs = 0.18–0.43$). Importantly, trait-level aggression as reported by adolescents was strongly correlated with higher carer reports of aggressive behaviour as measured in the SDQ, $r = 0.62$, $p < 0.001$, supporting the notion that our criterion measure was a valid indicator of aggressive behaviour.

## 3.2. Confirmatory analyses

### 3.2.1. Primary analysis

Linear regression modelling was used to evaluate the hypothesized linear and parabolic relationships between violent gaming and aggressive behaviour. Because research has shown that gender is robustly associated with both aggressive behaviour and violent game preference, the effects of violent gaming were evaluated holding variability linked to adolescent gender constant. To evaluate the links between violent gaming and aggression cited above, a regression model tested for a relationship between the two constructs. In the first step of this model, trait-level aggression scores reflecting physical, verbal, anger and hostility were entered along with participant gender. In the second step, the time spent playing violent video games was entered as a predictor. In the third step of this model, the parabolic term, the square of the time participants devote to violent video games, was entered. This product term represented a plausible nonlinear alternative pattern that might explain the relations between technology and youth outcomes as seen in other domains within technology use including games [2,67]. Of interest for this model was whether the linear and parabolic effects for violent gaming account for a significant share of variance in aggressive behaviour (table 3). Results from this model showed that neither the linear ($p = 0.402$) nor the parabolic ($p = 0.624$) predictors were statistically significant. In other words, these results did not support our prediction that there are statistically significant links relating violent gaming to adolescents' aggressive behaviour.

We have planned a more sensitive test of the potential harm of video game use by identifying a key inflection point for violent gaming effects on youth aggression. This was not possible because the parabolic term was not significant ($p = 0.624$), and as such we did not have an empirical basis to calculate the local extrema as planned. As a result, we did not split data at an inflection point value [68] or conduct the additional regression tests required to test if the nonlinear relationship between the time participants spends playing violent video games and aggression followed a step function like those found in other large-scale media effects research [2]. These results did not support the idea that there is a critical amount of violent video game play which serves as a tipping point for aggressive behaviour.

### 3.2.2. Equivalence testing

In order to know if the effect observed was practically significant, we directly compared the standardized semi-partial correlation coefficients to the best existing meta-analytic effect size estimate identified by Hilgard *et al.* [32], using the two one-sided tests procedure [69]. In this test, we contextualized the observed effect size estimate in terms of whether it is inferior (i.e. smaller), equivalent (i.e. falls within the same range as) or is superior (i.e. larger) to findings present in the existing literature [32], and in line with proposed minimum practical media effect sizes [70–72]. The semi-partial effect relating violent gaming to aggressive behaviour was $r = 0.01$. Further, we derived a 95% coincidence interval around this point estimate that ranged from $-0.08$ to $0.10$ using a bootstrapping approach with 10 000 iterations. Given this effect, $r = 0.01$ (95% CI $= -0.08$ to $0.10$), did not overlap with, and was clearly inferior to, $r = 0.21$ (95% CI $= 0.20–0.22$), we concluded this observed effect relating violent gaming to aggressive behaviour was both statistically and practically insignificant.

### 3.2.3. Sensitivity analysis

A second regression model examined the links between violent video game engagement and aggression using an alternative method for operationalizing violent game content using PEGI ratings. In line with a Stage 1 reviewer's suggestion, an additional piece of information, the PEGI age rating, was used to recode games with scores ranging from 0 to 3. A score of 0 was applied to games with no violence badge, a score of 1 was applied to games with a violence badge, a score of 2 was applied to games with a badge and an age rating of 16, and a score of 3 was applied to games with a badge and an age rating of 18. These

**Table 3.** Confirmatory hypothesis tests examining the relationship between adolescent's violent video game engagement and carer's ratings of adolescent's aggressive behaviour. The primary analysis uses an operationalization of violent video game engagement based on the amount of time participants devoted to games that PEGI has assigned a violent content badge. The sensitivity analysis adds information derived from PEGI age rating to operationalize violent game content.

| model | predictor variables | primary analysis | | | | | | sensitivity analysis | | | | | |
|---|---|---|---|---|---|---|---|---|---|---|---|---|---|
| | | unstandardized slopes | | | standardized slopes | | variance | unstandardized slopes | | | standardized slopes | | variance |
| | | b | 95% LL | 95% UL | $\beta$ | p | $R^2$ | b | 95% LL | 95% UL | $\beta$ | p | $R^2$ |
| Step 1 | gender | −0.021 | −0.235 | 0.194 | −0.005 | 0.849 | 0.000 | −0.013 | −0.260 | 0.234 | −0.003 | 0.918 | 0.000 |
| | physical aggression | 0.588 | 0.442 | 0.734 | 0.330 | 0.000 | 0.046 | 0.598 | 0.431 | 0.766 | 0.337 | 0.000 | 0.051 |
| | verbal aggression | 0.191 | −0.009 | 0.391 | 0.106 | 0.061 | 0.003 | 0.300 | 0.067 | 0.533 | 0.170 | 0.012 | 0.007 |
| | trait anger | 0.482 | 0.286 | 0.678 | 0.267 | 0.000 | 0.017 | 0.351 | 0.121 | 0.581 | 0.199 | 0.003 | 0.009 |
| | trait hostility | 0.070 | −0.086 | 0.226 | 0.038 | 0.379 | 0.001 | 0.045 | −0.132 | 0.222 | 0.025 | 0.619 | 0.000 |
| Step 2 | violent video game engagement (linear) | 0.009 | −0.037 | 0.056 | 0.011 | 0.688 | 0.000 | −0.001 | −0.021 | 0.019 | −0.004 | 0.899 | 0.000 |
| Step 3 | violent video game engagement (parabolic) | 0.002 | −0.007 | 0.011 | 0.028 | 0.624 | 0.000 | 0.000 | −0.002 | 0.001 | −0.013 | 0.851 | 0.000 |

ratings were combined with play time following the approach used to compute the dichotomous content coding and models, and followed the same procedure used in the primary regression analyses. Results from this model, presented in table 3, indicated both the linear ($p = 0.899$) and parabolic ($p = 0.851$) predictors were not statistically significant. These results did not support our prediction that there are statistically significant links relating violent gaming to adolescents' aggressive behaviour using the current operationalization of violent gaming.

## 3.3. Exploratory analyses

Two novel research questions which were not included in our Stage 1 submission came to mind after the manuscript was accepted, in principle, and data collection was completed. Both bear mention in this paper as they related to the extant literature and policy dimensions of the study of violent video game effects, and to the goals of this project. First, while games were being coded using PEGI ratings, we considered that we could also operationalize violent game content using game classifications from the North American market. Because North America is an important, but different, market than Europe for games, and the approach to rating games is different, we thought that a valuable addition to the analyses already conducted would be to seek convergent or divergent results by linking North American classifications to adolescents' aggressive behaviour. Second, while preparing analyses we recognized that an additional relevant report provided by adolescent participants' parents, namely, of their youngsters' prosocial behaviour, would provide an additional insight regarding the correlates of violent gaming. There is a growing literature on the links between gaming and prosocial behaviour and we decided the current data could speak to this literature as well [29,73,74].

### 3.3.1. North American operationalization of violent video game engagement

We examined links between gaming and aggression using ratings derived from North America's Entertainment Software Ratings Board (ESRB) in addition to the European system (i.e. PEGI) as defined in the primary analyses. Following the general approach used for PEGI ratings, games featuring more granular measures of blood, blood and gore, violence, fantasy violence, intense violence, sexual violence or references to violence were coded 1, whereas games without any of these content badges were coded 0. This coding was combined with play time estimates following the same approach used for PEGI-based violent gaming time estimates. A correlation analysis showed that participants' recent violent game time, using PEGI and ESRB ratings, were highly interrelated ($r = 0.80$, $p < 0.001$). Perhaps unsurprisingly then, results, presented in table 4, mirrored those observed in both the primary and sensitivity analyses. Violent game engagement was not a statistically significant linear ($p = 0.98$) or nonlinear ($p = 0.07$) predictor of aggressive behaviour.

### 3.3.2. Prosocial behaviour as an alternative outcome measure

A final series of models used the prosocial behaviours subscale of the SDQ to determine if carers' impressions of their child's helpfulness was influenced by violent video game play. To this end, carers rated the truth of five statements characterizing their adolescent as prosocial, including 'considerate of other people's feelings' and 'kind to younger children', using the same three-option scale used to evaluate aggressive behaviour. Scores on this five-item subscale were summed and ranged from 5 to 15 ($M = 12.08$, s.d. $= 2.29$, $\alpha = 0.77$). Following the approach used for aggressive behaviour, a hierarchical regression model tested for a relationship between the violent gaming and prosocial behaviour. Results from this analysis mirrored those for aggressive behaviour: there was no significant linear ($p = 0.49$) or nonlinear ($p = 0.70$) relation in evidence.

## 4. Discussion

The question of whether adolescent engagement with violent video games drives aggressive behaviour in young people is a critically important one. Indeed, our data indicated these games were regularly played by almost half of female and two-thirds of male teens in the UK. Given this popularity, one might argue that a small effect linking violent gaming to aggressive behaviour would have consequences for society as a whole [46]. To examine whether links can be evidenced, the present research applied the registered reports methodology to bring a novel and rigorous empirical lens to a scientific literature sharply divided on the effects of violent video games [29]. Our main interest

**Table 4.** Exploratory analyses examining the effects of examining the relationship between adolescent's violent video game engagement and carer's ratings of adolescent's aggressive behaviour and prosocial behaviour.

| outcomes | model | predictor variables | unstandardized slopes | | | standardized slopes | | variance |
|---|---|---|---|---|---|---|---|---|
| | | | b | 95% LL | 95% UL | β | p | R² |
| aggressive behaviour | Step 1 | gender | −0.021 | −0.235 | 0.194 | −0.005 | 0.849 | 0.000 |
| | | physical aggression | 0.588 | 0.442 | 0.734 | 0.330 | 0.000 | 0.046 |
| | | verbal aggression | 0.191 | −0.009 | 0.391 | 0.106 | 0.061 | 0.003 |
| | | trait anger | 0.482 | 0.286 | 0.678 | 0.267 | 0.000 | 0.017 |
| | | trait hostility | 0.070 | −0.086 | 0.226 | 0.038 | 0.379 | 0.001 |
| | Step 2 | violent video game engagement (linear) | −0.001 | −0.056 | 0.055 | −0.001 | 0.980 | 0.000 |
| | Step 3 | violent video game engagement (parabolic) | 0.011 | −0.001 | 0.023 | 0.104 | 0.067 | 0.003 |
| prosocial behaviour | Step 1 | gender | 0.660 | 0.350 | 0.969 | 0.144 | 0.000 | 0.020 |
| | | physical aggression | −0.340 | −0.551 | −0.129 | −0.166 | 0.002 | 0.011 |
| | | verbal aggression | −0.239 | −0.527 | 0.049 | −0.115 | 0.104 | 0.003 |
| | | trait anger | −0.122 | −0.404 | 0.161 | −0.059 | 0.398 | 0.001 |
| | | trait hostility | −0.116 | −0.341 | 0.110 | −0.055 | 0.315 | 0.001 |
| | Step 2 | violent video game engagement (linear) | 0.023 | −0.043 | 0.090 | 0.024 | 0.493 | 0.001 |
| | Step 3 | violent video game engagement (parabolic) | 0.002 | −0.010 | 0.015 | 0.028 | 0.701 | 0.000 |

concerned the relationship between the amount of violent video game play teens engaged in the previous month and the extent to which their parents judged their behaviour as aggressive during this time. In line with this goal, we evaluated a number of confirmatory and exploratory models that tested the prediction that higher levels of engagement with violent games would be positively associated with more aggressive behaviour and less prosocial behaviour in young people. Broadly speaking, findings from our study provided evidence that this was not the case. Said differently, the results derived from our hypothesis testing did not support the position that violent gaming relates to aggressive behaviour.

In order to contextualize empirical tests within current debates, we based our study design and analysis plan on the most recent and comprehensive synthesis of the existing literature base [32]. Our aim was to empirically observe and extend the basic idea at the heart of the violent video game literature, namely that exposure to violence in gaming contexts could have a carry-over effect which influences the extent to which aggressive behaviours are exhibited in everyday life. At the same time, we wanted to eliminate sources of bias which could be expected to influence the quality of inferences one might draw about the effects of video games. To this end, we considered some of the most prevalent pitfalls present in the existing literature and took active steps to account for these issues in the study design.

First, we observed that many researchers ask participants not only to estimate their own aggressive behaviours but also to provide subjective ratings of the violence present in the games they play. This commonly used method requires participants to subjectively judge both the predictor (game violence level) and criterion (their aggression level) constructs, thereby introducing a number of potential confounds. For example, it may be that more aggressive young people tend to rate the games they play as having more aggression because underlying differences in aggressive or hostile perceptual biases orient them to these game features, whereas less aggressive young people attend to different aspects of play. Alternatively, youngsters who are willing to report in less desirable ways might be more likely to report on both their own and in-game aggression. In the present study, we minimized this source of bias in the predictor by having an independent coder classify the content of games using European and North American rating systems. Additionally, we relied on carers, not adolescents, to judge the presence or the absence of aggressive behaviour. We took these steps to minimize the chance that self-reporting biases or common method variance would inflate or influence the study's estimates of the correlations between behaviours in gaming and real-world contexts.

Second, our review of the literature made it clear that there is a high level of methodological flexibility in the ways that aggression outcomes and violent gaming are assessed [46,75]. As noted previously, in at least one dataset (https://osf.io/3gdt5/), three distinct sets of variables drawn from eight Likert-style judgements about games have been used in different combinations in different papers. This method of selectively operationalizing fundamental theoretical constructs undermines our confidence in the inferences we might draw about gaming effects. Our study, then, contributes to this literature base, as we prespecified exactly how we would operationalize key variables before conducting the study. By doing so, the research provides a template that media effects researchers could follow when adopting a hypothesis testing approach with other important and plausible forms of technology influence.

Finally, all studies of violent video game effects we have encountered use statistical significance as a surrogate for determining if the effects of gaming are practically significant in real-world terms. The current study framing diverged from this practice in that before conducting the study, we set both a criterion for statistical significance and an *a priori* threshold informed by the extant literature as a criterion for practical significance. Findings can be interpreted with greater confidence because these standards were set for type 1 and 2 error control and the sample was sufficiently large for a fair and sensitive test of the null and alternative hypotheses. Additionally, because we prespecified how we would interpret effects larger, smaller or falling in the same range as estimates from meta-analysis [32], the current work provides information regarding whether these links are robust enough to be considered evidence for an effect by parents, policy-makers and professional organizations.

We believe the current work is the second study in the area to set the number of observations based on an *a priori* power analysis [47], and the first to specify a minimum effect size of theoretical interest. In examining media effects, researchers focus on a number of topics including emotional contagion in social networks [76], technology addiction and psychological well-being in digital contexts [2], where this approach would be preferable to using statistical significance as the only arbiter of true effects. Indeed, statistical significance does not necessitate a subjective state that humans can distinguish on a personal level [77,78]. Given interactive media and their effects are inherently subjective, we believe working toward adoption of a new standard for interpreting the practical significance of media effects would serve the literature well as a benchmark against which putative effects of emergent technologies such as virtual reality, augmented reality and artificial intelligence may be judged.

Keeping in mind the steps taken to ensure methodological rigour, the current results bear directly on the contentious literature surrounding games. Results from these confirmatory analyses provided evidence that adolescents' recent violent video game play is not a statistically or practically significant correlate of their aggressive behaviour as judged by carers. Preregistered sensitivity and exploratory analyses demonstrated this finding was consistent across three different operationalizations of violent game content and two ways of measuring key adolescent behaviours relating to aggressive and prosocial behaviours. In other words, we found adolescents were not more or less likely to engage in aggressive or prosocial behaviours as a function of the amount of time they devoted to playing violent games. This pattern of findings further suggests that links reported in the literature might be influenced by publication bias, selective reporting, or an artefact of unobserved or hidden moderators, as has been previously suspected [45,78]. We argue that this study speaks to the key question of whether adolescents' violent video game play has a measurable effect on real-world aggressive behaviour. On the basis of our evidence, the answer is no. This is *not* to say that some mechanics and situations in gaming do not foment angry feelings or reactions in players such as feelings of incompetence [31], trash talking [79] or competition [80]. These topics provide promising avenues for inquiry and have direct implications for literature focused on antisocial behaviours such as bullying, trolling and griefing [81,82]. Instead, we argue that mere exposure to, and enactment of, putatively violent virtual acts in gaming contexts in aggregate is unlikely, on its own, to bear positively on perceivable differences in adolescents' aggression in real-world settings.

Further, this is not to say that we could rule out a correlation between every operationalization of gaming and every measure of adolescents' violent behaviour. In fact, we observed at least three such correlations ranging in size from when considering uncorrected models using variables we did not hypothesize would be correlated with aggressive behaviour. For example, our positive control check question, a simple 5-point Likert scale that asked participants to rate their own intensity of engagement with games happened to be positively correlated ($r = 0.25$) to aggressive behaviour. If we had not preregistered our empirical approach and felt motivated to publish a positive result we might have seized on this correlation and made it the central focus of our research report. This possibility, capitalizing on chance, underlines the value of the registered reports framework for documenting the hypothesis generation and testing process and formalized data-driven approaches for exploratory data analysis such as specification curve analysis [83]. In isolation, a cherry-picked result such as this might add undue weight to the moral panic surrounding electronic gaming. Study preregistration and registered reports act as bulwark against drawing such *post hoc* inferences.

With that understood, the work does have limitations that constitute concrete paths for those seeking to extend the robustness of the inferences that can be drawn about the effects of violent video game engagement. First, our work is based entirely on self-report data. A number of recent investigations have integrated approaches combining user and trace data with time-series analyses to draw community-level (versus individual) inferences about the effects of gaming on public reports of antisocial behaviours [84]. In our view, these kinds of data, principally held by gaming companies themselves, would provide an invaluable resource if linked with existing large-scale datasets such as the British Household Panel Study [85]. This could provide a context to understand gaming effects set against a rich data milieu, including information on objective gaming behaviours, social, familial, school, individual and genetic level factors. Second, the present study is based on cross-sectional data. Although findings derived from preregistered experimental research are in line with laboratory-based theory-testing research on gaming effects [47], it remains an open question whether play has an enduring or compounding effect on aggressive behaviour over time. It is possible that the retrospective month-long 'snap shot' this study uses missed a critical dynamic which develops over repeated engagement with video games. To this end, field or natural experiments using multi-wave random-intercepts cross-lagged panel modelling should follow the present work to provide an expanded test of the effects of violent gaming on human behaviour. Finally, in this study, we draw general inferences about gaming effects in a general way across the population as a whole. It might be the case that specific cohorts of people sharing background factors associated with technology use such as carer educational attainment or material deprivations are more or less likely to be influenced by their experiences with virtual environments [86]. If indeed this is the case, such groups should be the focus of targeted programmes of exploratory and confirmatory research. Findings derived from such analyses would enable evidence-based interventions, meaningful professional guidance and productive policy-making. Until such findings are confirmed, however, we strongly would caution about drawing impulsive, thoughtless or potentially stigmatizing conclusions about members of such groups [76].

## 4.1. Closing remarks

Despite the null findings identified in the present study, history gives us reason to suspect the idea that violent video games drives aggressive behaviour will remain an unsettled question for parents, pundits and policy-makers. Although our results do have implications for these stakeholders, the present work holds special significance for those studying technology effects, in general, and video games, in particular. It is crucial that scientists conduct work with openness and rigour if we are to build a real understanding of the positive and negative dynamics and impacts of technology in people's lives [76]. This is among the first studies to test the effects of violent gaming on human aggression using a preregistered hypothesis-testing framework and the first to do so following the registered reports protocol. The results provide confirmatory evidence that violent video game engagement, on balance, is not associated with observable variability in adolescents' aggressive behaviour. A healthy ecosystem of exploratory and registered research reports will enable scientists to conduct metanalytic research to evaluate the inferences drawn from these methodologies. Only then will we be able to examine the pathways by which aggressive play might relate to real-world aggression in novel, incremental and empirically robust ways. With this evidence in hand, we will be able to judge if the attention and resources allocated to this topic, spent at the expense of other important questions of the digital age, is empirically justified.

Ethics. The research involves humans and the protocol for this study has been reviewed and approved by the Central University Ethics Committee of the University of Oxford (C1A17023).
Data accessibility. All materials, code and data are available for download on the Open Science Framework (https://osf.io/rkw6z/).
Authors' contributions. A.K.P. and N.W. made substantial contributions to conception and design of the study and collected, analysed, and interpreted the data; drafted and revised the Stage 1 and Stage 2 versions of the paper. Both give approval of the version to be published and are accountable for all aspects of the work in ensuring that questions related to the accuracy or integrity of any part of the work can be appropriately investigated and resolved.
Competing interests. We have no competing interests.
Funding. Funding for this research is provided by the John Fell Fund (163/079).
Acknowledgements. No one contributed to the study but did not meet the authorship criteria.

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
