## [Reviewer comments · Royal Society Open Science]

Review History

RSOS-170903.R0 (Original submission)

Review form: Reviewer 1 (Christopher Ferguson)

Is the language acceptable?

Yes

Do you have any ethical concerns with this paper?

No

Have you any concerns about statistical analyses in this paper?

No

Recommendation?

Accept with minor revision

Comments to the Author(s)

I enjoyed reading this preregistered replication proposal for RSOS. The proposal is well written, covers the prior literature accurately, and makes a good case for their study. The authors should be congratulated for using preregistered designs in their analyses.

I do, however, have a few comments that might improve their design.

First, as a minor note, on page 2 the authors conclude that Hilgard et al., report an effect size for video games on aggression as $r = .21$. I don't think that's exactly correct, looking at Table 3 from the Hilgard paper. From Hilgard, adjusted effect size estimates vary depending upon correction method used, so I'm not sure you can point to a single adjustment approach. I'm guessing that the authors here may be referring to the "full" studies for aggressive behavior in correlational methods which is indeed .21 (but confidence interval is .12, .28) using PET. You might just want to be clearer exactly where that effect size is coming from as this confused me a bit. Effect sizes were much lower for experimental studies, but that may not be relevant here for a correlational study.

As to the design, I had a few comments. First, I would *strongly* urge the authors to include control variables identified as important in the literature. This includes not just gender (which is, indeed, important), but also mental health (particularly depression), personality (trait aggression), family environment (psychological abusiveness between parents has been observed as important in prior studies), and peer delinquency. This could be achieved with just a few additional variables. At present, the current design's main weakness is the third-variable problem. The current design is too basic. But I think this will be easy for the authors to fix.

Also, instead of separate regression models for boys and girls, I would simply include gender as a dummy coded variable in the regression model.

Third, I know this would eat up more time, but the current study is also "demand characteristic" ...although I grant having parents and kids fill out different forms helps some. Assuming that the parents and kids are both online taking the survey semi-together, the potential for DC remains high though. I'd suggest including some distractor questions to make the hypotheses of the study less obvious, particularly to parents who may overrespond to aggression once they see their kids responding to video game questionnaires. I'd ask the kids about other hobbies, plus some other things, and ask the parents some irrelevant stuff about their kids too.

Lastly, I encourage the authors to predetermine effect size cut-offs for practical significance. Particularly with such a large sample, there's the potential for getting "statistically significant" results that are nonetheless trivial...or due to some of the noise in the study (like demand characteristics!)...not real effects. An $r = .10$ would be a bare minimum cut-off for interpretation, with .20 probably being better if you want to talk about clinically significant effects.

I hope these comments are helpful. I do think this would be an important study, particularly if we get a few more control variables in there!

Signed,
Chris Ferguson
Stetson University

Review form: Reviewer 2

Is the language acceptable?

Yes

Do you have any ethical concerns with this paper?

No

Have you any concerns about statistical analyses in this paper?

No

Recommendation?

Accept with minor revision

Comments to the Author(s)

RSOS-170903

I have now had an opportunity to review the proposed study "Violent video game engagement and adolescents' aggressive behavior: A registered report" for possible publication in Royal Society Open Science. Consistent with RSOS reviewing guidelines, I have addressed the following items in my review.

The significance of the research question

The significance of the proposed research question is undoubtedly important. The relationship between exposure to violent media (with violent video games being one specific type of media) and aggression has theoretical and societal relevance. And, unfortunately, the extant research into this effect has left researchers with an uncertain empirical foundation to build upon (indeed, some people believe this empirical foundation is strong and some people believe it is a complete disaster). This has left many individuals with the impression that we need to (re?)establish even the most basic effects relevant to this hypothesis: Is there even a bi-variate association between playing violent video games and aggression? In my opinion, this means that we need pre-registered and "open" studies such as this one even if it seems to rehash effects that some individuals believe are well-established.

In making the case for the significance of the research question, the authors should cite the recent article by Anderson et al. (2017).

Anderson, C. A., Suzuki, K., Swing, E. L., Groves, C. L., Gentile, D. A., Prot, S., ... & Jelic, M. (2017). Media Violence and Other Aggression Risk Factors in Seven Nations. *Personality and Social Psychology Bulletin*, 43(7), 986-998.

The logic, rationale, and plausibility of the proposed hypotheses

The logic and rationale of the proposed hypotheses are not well articulated. I believe that a lot more could be done to argue why it is plausible that exposure to violent video games could cause subsequent aggression. The current Introduction discusses relevant meta-analyses and general criticisms of this area of research; however, there is little discussion about the theoretical underpinnings of the stated hypothesis. This would probably mean discussing hypotheses that are derived from social-cognitive models of aggression (e.g., exposure to violent video games reinforces aggressive responding/desensitizes participants to violence, etc.). Also, although I am sure the authors intended to be skeptical and critical, the Introduction seems to come off as a bit biased against the existence of the effect. I really dislike making vague statements about authors' writing style, but there does seem to be a subtle negative tone that seeps into the Introduction.

I can imagine that some people may quibble with the fact that the primary hypothesis is *merely* a correlation and believe that *more sophisticated* moderators of this believed-to-be-well-established effect should be tested, but I believe the simplicity of the hypothesis is also the strength of the hypotheses. Although it is true that moderators may strengthen or weaken the relationship between violent video game exposure and aggression, the cited meta-analyses

clearly argue the simple bi-variate relationship between violent video game exposure and aggression would be expected.

My only comment about the conceptual hypotheses is that the authors imply, but do not explicitly state, the direction of the expected relationship. The authors should explicitly state they expect a *positive* relationship between violent video game exposure and aggression. Also, the authors should specify the expected direction of the parabolic relationship (i.e., does the relationship strengthen or weaken as violent video game increases?). I could post-hoc spin a plausible sounding explanation for either direction of a parabolic relationship (if it emerged).

My major criticism is not in the hypotheses per se, but in the operationalization of the hypotheses. The outcome measure is the *Conduct Problems* subscale of the SDQ. To my read, there is only one item that actually measures aggression (i.e., Often fights with other children or bullies them). The other items seem to ask about non-aggressive conduct problems (e.g., often has temper tantrums or hot tempers; often lies or cheats; steals from home, school, or elsewhere). This is the same shortcoming as the Anderson et al. (2017) paper mentioned above that also claimed to measure aggressive behaviors. However, these items do NOT measure aggressive behaviors and, thus, are inappropriate for testing the hypotheses. This is a big, big deal. Is it too late to include parent reports of their child's aggressive behaviors? Or is it possible that the authors can modify their hypotheses to state they are interested in conduct problems in general, not aggression specifically?

A minor point is the framing of the timeframe of the items. The child respondents will be self-reporting on their video game play in the past month and there does not seem to be a timeframe for parents' reports of their children's behaviors. Is it possible to tell parents to think about their child's behavior in the past month? Or justify why you would not want to instruct the children and parents to think of the same timeframe when completing the study.

The soundness and feasibility of the methodology and analysis pipeline/ Clarity and degree of methodological detail

This is a fairly straightforward design so I do not have any problems with the methods. The only aspect that gives me slight pause is that the methods rely heavily on the ability of the research firm to recruit participants. Data collection could be difficult because the unit of analysis is actually a child-parent dyad, which means that you could have issues with some dyads missing data from one person (see my comment below about pre-registering a plan to handle cases where one person in the dyad does not provide a response). Also, is there anything in place to ensure that respondents do not influence each other? For example, children and parents who complete the survey independently will likely give different responses than children and parents who sit down and complete the survey jointly. What instructions will children/parents be given when they are asked to participate? Will they be told to keep their responses confidential?

Can the authors add more detail about how participants and their parents will be sampled? How will the research firm find and recruit participants? How will participants be compensated?

Also, awesome that this will be a representative sample. Good job!!

Sufficiently clear and detailed description of the methods to prevent undisclosed flexibility in the experimental procedures of analysis pipeline

I would like to see information about the following aspects of the analysis pipeline. First, are there any exclusion criteria that will be considered. For example, if participants provide missing data, will the authors omit those participants or will the authors attempt to impute the missing values? If the latter, how will the authors impute data?

Also, is the relationship between children and parents one-to-one? For example, is it possible to have more than one child respondent who share the same parent? How will the authors handle cases of children nested within parents? Randomly select one child? Take the oldest/youngest child? Analyze the data using multi-level modeling?

Similarly, what happens if only a child or only a parent responds (effectively making some units of analysis to have missing predictor variables or missing outcome variables)? Will those participants be completely omitted from analyses?

Second, please have the authors justify why they will conduct two parallel analyses separately for males and females. From my perspective, it would seem that you would merely enter the child gender and the interaction term as a moderator in one common analysis. If the authors ultimately decide to conduct one common analysis with gender as a moderator, the authors also should specify how they will code for child gender (e.g., dummy coding where 0=female and 1=male, etc.).

Third, is there any other measures from the data collection process that could be used as “control” variables? For example, because this is a correlational method, it is plausible that there is a third-variable problem where an unmeasured variable (e.g., trait aggression) causes both exposure to violent video games and aggression. Now would be a good time to add those variables (if any) into planned analyses.

Is the exploratory analysis necessary? It seems to be merely a different way to test the primary hypothesis.

Have authors considered sufficient outcome-neutral conditions to test the stated hypotheses?

Is it possible to add a “positive control” or some instructional manipulation checks? For example, can the authors add an item like “I spend a lot of time playing video games” and see if this item positively correlates with their self-reported amount of time playing video games. Although it is possible to have a divergence between objective measures of time spent in an activity (e.g., hours per week) and a subjective measure (e.g., I *feel* like I spend a lot of time doing an activity), I would put a high prior on observing a positive relationship between the two items. Something like this would demonstrate that the researchers are able to detect predictable relationships between variables with their data collection procedure, which would help interpret results of the tests of the primary hypotheses.

In the Introduction the authors discuss the possibility of “mischievous responding”, but also don’t seem to address this problem beyond avoiding child self-reports. There is still issues about participants’ (both parents and children) attending to the questions, responding honestly, etc. that would seem to be important to consider here. If anything is added to potentially exclude participants who give bad data, there would need to be decisions made about how those exclusion criteria would be enforced (e.g., are dyads excluded if either the child or the parent provides bad data?).

In summary, assuming the authors would be able to add/modify the outcome variable to measure actual aggressive behaviors, I strongly support this project moving forward. These types of projects are critical to advancing our understanding of the effects of violent video game exposure and aggression.

Review form: Reviewer 3 (Peter Etchells)

Is the language acceptable?

Yes

Do you have any ethical concerns with this paper?

No

Have you any concerns about statistical analyses in this paper?

No

Recommendation?

Accept with minor revision

Comments to the Author(s)

The authors present a proposal for a well-powered study assessing whether there is any association between violent content in video games and recent aggressive behaviour in adolescents. The rationale for the study is sound, and the study would present a substantial advance in the field.

The study as presented is sufficiently clear to allow for an exact replication, and the authors have presented a fair consideration of sufficient outcome-neutral conditions. The analytic methods suggested are appropriate, and have been used in similar recent studies on screen time and wellbeing.

In general, I think that this will be a much-needed addition to the research literature on violent video game effects, a field that desperately needs preregistered studies so as to protect against undisclosed flexibility in analytic techniques (as the authors note in their introduction). We still know very little about how violent game content has an impact, if any, on aggressive behaviour. As such, the present study has the potential to drive forward a new way of thinking about video games effects research.

By and large, the authors present a strong preregistration plan. I do think the study needs some additional tweaks, however.

1) Given that the study is explicitly looking at short-term variations in aggressive behaviour, it would be useful to have some sort of baseline measure of pre-existing aggression in the adolescent sample. I appreciate that this is difficult, given the cross-sectional nature of the design, but it is a limitation that certainly needs to be addressed in some way. I wonder whether use of something like the trait component of the state-trait anger expression inventory (STAXI) would help in this regard?

2) I'm torn on the video game violence measure. In a way, it is a very elegant design – game violence is usually poorly operationalised in these sorts of studies, and by using PEGI classifications as a basis, the authors are tying the violence categorisation to something that makes realistic sense in terms of parental game choices. On the other hand, I'm slightly worried that coding game violence as either 1 (violent) or 0 (not-violent) is too broad, and might have the potential to lose important information about specific game content that could be driving any associations with aggressive behaviour. For example, under the PEGI system, both World of Warcraft and Call of Duty are classed as 'violent' games. In reality, the way in which in-game violence plays out in these games is very different. I wonder, as a way to get over this issue while still retaining the elegance of using PEGI ratings, whether a measure of violence could be developed that takes into account the PEGI age rating? To go back to my example, World of Warcraft is PEGI 12, whereas Call of Duty is PEGI 18. In that sense then, they are categorised as different levels of violence.

As a simple tweak then, would it be appropriate to have 4 levels of the explanatory variable: 0 (no violence badge); 1 (violence badge, 12 rating); 2 (violence badge, 16 rating); 3 (violence badge, 18 rating)? This might also have the advantage of considering age-appropriateness as a potential factor; the proposed adolescent group will be aged 14-15, so in theory shouldn't be playing 16 or 18-rated games.

I look forward to further correspondence from the authors.

Signed,
Dr Pete Etchells

Review form: Reviewer 4 (Adam Lobel)

Is the language acceptable?

Yes

Do you have any ethical concerns with this paper?

No

Have you any concerns about statistical analyses in this paper?

No

Recommendation?

Accept in principle

Comments to the Author(s)

Dear Dr. Przybylski and Dr. Weinstein,

It is a pleasure to be able to review this project and to be given the opportunity to engage with you at this stage. My compliments on formulating a concise, cogent, and much-needed call for the study you are proposing. Studies of the scale and methodological rigor you present are much needed, and it is encouraging to see such competent researchers rise to the challenge.

I was left wondering a couple of things regarding your study:

1) The literature on violent video gaming has recently expanded into examining the influence of cooperative and competitive gaming. Given the proposed size of your study's sample, I think you are in the unique position to be able to add clarity and great value by including these as potential mediating variables. Was this a consideration of yours? And if so, what influenced your decision not to investigate these variables? (Related to this, your study already asks participants whether the game they have been playing was being played in single or multiplayer mode.)

2) I commend your choice to rely on the PEGI ratings. I wonder whether you think it is important to distinguish between different types of violence ratings applied by PEGI (e.g. "cartoon violence" vs. "realistic violence").

I also have a recommendation for your methods:

Based on past studies conducted by myself and my colleagues, you may want to ask for more specificity when asking participants to nominate the games they played most over the past month. While devoted gamers are likely to be very specific about the games they played, more casual gamers are apt to list generic titles (e.g. "Mario") or to simply list the name of websites that host large numbers of simple Flash games and mini-games. You may therefore wish to give your participants a bit more directive with this item. I also miss the wording for how you will ask participants to describe whether they play single player or multiplayer (described as "variable 4" on page 5, but not listed in Appendix B).

Finally, some suggestions for very minor wording tweaks:

-Page 1, Paragraph 1. "Following preregistered analysis plan multiple regression analyses will test..." Add a comma after "plan".

-Page 4, Paragraph 3. "This survey-based study will... measure youth aggression with caregiver responses to a widely used in behavioural screening questionnaire..." I think this would read better with the word "questionnaire" moved to after "used".

Thank you for your work,
Best of luck,
Adam Lobel, PhD

Decision letter (RSOS-170903.R0)

07-Aug-2017

Dear Dr Przybylski,

The Editors assigned to your Stage 1 Registered Report ("Violent Video Game Engagement and Adolescents' Aggressive Behaviour: A Registered Report") have now received comments from reviewers. We would like you to revise your paper in accordance with the referee and editors suggestions which can be found below (not including confidential reports to the Editor). Please note this decision does not guarantee eventual acceptance.

Please submit a copy of your revised paper within three weeks (i.e. by the 29-Aug-2017). If deemed necessary by the Editors, your manuscript will be sent back to one or more of the original reviewers for assessment.

Kind regards,
Alice Power
Editorial Coordinator
Royal Society Open Science

on behalf of Chris Chambers
Registered Reports Editor, Royal Society Open Science
openscience@royalsociety.org

Associate Editor Comments to Author:

Comments to the Author:

Four expert reviewers have now assessed the manuscript. All are positive but offer a range of critical suggestions that cover the full spectrum of Stage 1 review criteria. Two issues in particular have been highlighted by the editorial team. First, as noted by Reviewer 1, the authors should consider the inclusion of control variables. Second, Reviewer 2 makes the important point that the outcome measure does not provide a very precise measure of aggression, and indeed mostly captures non-aggressive conduct. Please be sure to address these two major points in addition to the full range of detailed comments raised by all reviewers.

Comments to Author:

Reviewer: 1

Comments to the Author(s)

I enjoyed reading this preregistered replication proposal for RSOS. The proposal is well written, covers the prior literature accurately, and makes a good case for their study. The authors should be congratulated for using preregistered designs in their analyses.

I do, however, have a few comments that might improve their design.

First, as a minor note, on page 2 the authors conclude that Hilgard et al., report an effect size for video games on aggression as $r = .21$. I don't think that's exactly correct, looking at Table 3 from the Hilgard paper. From Hilgard, adjusted effect size estimates vary depending upon correction method used, so I'm not sure you can point to a single adjustment approach. I'm guessing that the authors here may be referring to the "full" studies for aggressive behavior in correlational methods which is indeed .21 (but confidence interval is .12, .28) using PET. You might just want to be clearer exactly where that effect size is coming from as this confused me a bit. Effect sizes were much lower for experimental studies, but that may not be relevant here for a correlational study.

As to the design, I had a few comments. First, I would *strongly* urge the authors to include control variables identified as important in the literature. This includes not just gender (which is, indeed, important), but also mental health (particularly depression), personality (trait aggression), family environment (psychological abusiveness between parents has been observed as important in prior studies), and peer delinquency. This could be achieved with just a few additional variables. At present, the current design's main weakness is the third-variable problem. The current design is too basic. But I think this will be easy for the authors to fix.

Also, instead of separate regression models for boys and girls, I would simply include gender as a dummy coded variable in the regression model.

Third, I know this would eat up more time, but the current study is also "demand characteristic" ...although I grant having parents and kids fill out different forms helps some. Assuming that the parents and kids are both online taking the survey semi-together, the potential for DC remains high though. I'd suggest including some distractor questions to make the hypotheses of the study less obvious, particularly to parents who may overrespond to aggression once they see their kids responding to video game questionnaires. I'd ask the kids about other hobbies, plus some other things, and ask the parents some irrelevant stuff about their kids too.

Lastly, I encourage the authors to predetermine effect size cut-offs for practical significance. Particularly with such a large sample, there's the potential for getting "statistically significant"

results that are nonetheless trivial...or due to some of the noise in the study (like demand characteristics!)...not real effects. An $r = .10$ would be a bare minimum cut-off for interpretation, with $.20$ probably being better if you want to talk about clinically significant effects.

I hope these comments are helpful. I do think this would be an important study, particularly if we get a few more control variables in there!

Signed,
Chris Ferguson
Stetson University

Reviewer: 2

Comments to the Author(s)

RSOS-170903

I have now had an opportunity to review the proposed study "Violent video game engagement and adolescents' aggressive behavior: A registered report" for possible publication in Royal Society Open Science. Consistent with RSOS reviewing guidelines, I have addressed the following items in my review.

The significance of the research question

The significance of the proposed research question is undoubtedly important. The relationship between exposure to violent media (with violent video games being one specific type of media) and aggression has theoretical and societal relevance. And, unfortunately, the extant research into this effect has left researchers with an uncertain empirical foundation to build upon (indeed, some people believe this empirical foundation is strong and some people believe it is a complete disaster). This has left many individuals with the impression that we need to (re?)establish even the most basic effects relevant to this hypothesis: Is there even a bi-variate association between playing violent video games and aggression? In my opinion, this means that we need pre-registered and "open" studies such as this one even if it seems to rehash effects that some individuals believe are well-established.

In making the case for the significance of the research question, the authors should cite the recent article by Anderson et al. (2017).

Anderson, C. A., Suzuki, K., Swing, E. L., Groves, C. L., Gentile, D. A., Prot, S., ... & Jelic, M. (2017). Media Violence and Other Aggression Risk Factors in Seven Nations. *Personality and Social Psychology Bulletin*, 43(7), 986-998.

The logic, rationale, and plausibility of the proposed hypotheses

The logic and rationale of the proposed hypotheses are not well articulated. I believe that a lot more could be done to argue why it is plausible that exposure to violent video games could cause subsequent aggression. The current Introduction discusses relevant meta-analyses and general criticisms of this area of research; however, there is little discussion about the theoretical underpinnings of the stated hypothesis. This would probably mean discussing hypotheses that are derived from social-cognitive models of aggression (e.g., exposure to violent video games reinforces aggressive responding/desensitizes participants to violence, etc.). Also, although I am sure the authors intended to be skeptical and critical, the Introduction seems to come off as a bit biased against the existence of the effect. I really dislike making vague statements about authors' writing style, but there does seem to be a subtle negative tone that seeps into the Introduction. I can imagine that some people may quibble with the fact that the primary hypothesis is *merely* a correlation and believe that *more sophisticated* moderators of this believed-to-be-well-established effect should be tested, but I believe the simplicity of the hypothesis is also the strength of the hypotheses. Although it is true that moderators may strengthen or weaken the relationship between violent video game exposure and aggression, the cited meta-analyses

clearly argue the simple bi-variate relationship between violent video game exposure and aggression would be expected.

My only comment about the conceptual hypotheses is that the authors imply, but do not explicitly state, the direction of the expected relationship. The authors should explicitly state they expect a *positive* relationship between violent video game exposure and aggression. Also, the authors should specify the expected direction of the parabolic relationship (i.e., does the relationship strengthen or weaken as violent video game increases?). I could post-hoc spin a plausible sounding explanation for either direction of a parabolic relationship (if it emerged).

My major criticism is not in the hypotheses per se, but in the operationalization of the hypotheses. The outcome measure is the *Conduct Problems* subscale of the SDQ. To my read, there is only one item that actually measures aggression (i.e., Often fights with other children or bullies them). The other items seem to ask about non-aggressive conduct problems (e.g., often has temper tantrums or hot tempers; often lies or cheats; steals from home, school, or elsewhere). This is the same shortcoming as the Anderson et al. (2017) paper mentioned above that also claimed to measure aggressive behaviors. However, these items do NOT measure aggressive behaviors and, thus, are inappropriate for testing the hypotheses. This is a big, big deal. Is it too late to include parent reports of their child's aggressive behaviors? Or is it possible that the authors can modify their hypotheses to state they are interested in conduct problems in general, not aggression specifically?

A minor point is the framing of the timeframe of the items. The child respondents will be self-reporting on their video game play in the past month and there does not seem to be a timeframe for parents' reports of their children's behaviors. Is it possible to tell parents to think about their child's behavior in the past month? Or justify why you would not want to instruct the children and parents to think of the same timeframe when completing the study.

The soundness and feasibility of the methodology and analysis pipeline/ Clarity and degree of methodological detail

This is a fairly straightforward design so I do not have any problems with the methods. The only aspect that gives me slight pause is that the methods rely heavily on the ability of the research firm to recruit participants. Data collection could be difficult because the unit of analysis is actually a child-parent dyad, which means that you could have issues with some dyads missing data from one person (see my comment below about pre-registering a plan to handle cases where one person in the dyad does not provide a response). Also, is there anything in place to ensure that respondents do not influence each other? For example, children and parents who complete the survey independently will likely give different responses than children and parents who sit down and complete the survey jointly. What instructions will children/parents be given when they are asked to participate? Will they be told to keep their responses confidential?

Can the authors add more detail about how participants and their parents will be sampled? How will the research firm find and recruit participants? How will participants be compensated?

Also, awesome that this will be a representative sample. Good job!!

Sufficiently clear and detailed description of the methods to prevent undisclosed flexibility in the experimental procedures of analysis pipeline

I would like to see information about the following aspects of the analysis pipeline. First, are there any exclusion criteria that will be considered. For example, if participants provide missing data, will the authors omit those participants or will the authors attempt to impute the missing values? If the latter, how will the authors impute data?

Also, is the relationship between children and parents one-to-one? For example, is it possible to have more than one child respondent who share the same parent? How will the authors handle cases of children nested within parents? Randomly select one child? Take the oldest/youngest child? Analyze the data using multi-level modeling?

Similarly, what happens if only a child or only a parent responds (effectively making some units of analysis to have missing predictor variables or missing outcome variables)? Will those participants be completely omitted from analyses?

Second, please have the authors justify why they will conduct two parallel analyses separately for males and females. From my perspective, it would seem that you would merely enter the child gender and the interaction term as a moderator in one common analysis. If the authors ultimately decide to conduct one common analysis with gender as a moderator, the authors also should specify how they will code for child gender (e.g., dummy coding where 0=female and 1=male, etc.).

Third, is there any other measures from the data collection process that could be used as “control” variables? For example, because this is a correlational method, it is plausible that there is a third-variable problem where an unmeasured variable (e.g., trait aggression) causes both exposure to violent video games and aggression. Now would be a good time to add those variables (if any) into planned analyses.

Is the exploratory analysis necessary? It seems to be merely a different way to test the primary hypothesis.

Have authors considered sufficient outcome-neutral conditions to test the stated hypotheses?

Is it possible to add a “positive control” or some instructional manipulation checks? For example, can the authors add an item like “I spend a lot of time playing video games” and see if this item positively correlates with their self-reported amount of time playing video games. Although it is possible to have a divergence between objective measures of time spent in an activity (e.g., hours per week) and a subjective measure (e.g., I *feel* like I spend a lot of time doing an activity), I would put a high prior on observing a positive relationship between the two items. Something like this would demonstrate that the researchers are able to detect predictable relationships between variables with their data collection procedure, which would help interpret results of the tests of the primary hypotheses.

In the Introduction the authors discuss the possibility of “mischievous responding”, but also don’t seem to address this problem beyond avoiding child self-reports. There is still issues about participants’ (both parents and children) attending to the questions, responding honestly, etc. that would seem to be important to consider here. If anything is added to potentially exclude participants who give bad data, there would need to be decisions made about how those exclusion criteria would be enforced (e.g., are dyads excluded if either the child or the parent provides bad data?).

In summary, assuming the authors would be able to add/modify the outcome variable to measure actual aggressive behaviors, I strongly support this project moving forward. These types of projects are critical to advancing our understanding of the effects of violent video game exposure and aggression.

Reviewer: 3

Comments to the Author(s)

The authors present a proposal for a well-powered study assessing whether there is any association between violent content in video games and recent aggressive behaviour in adolescents. The rationale for the study is sound, and the study would present a substantial advance in the field.

The study as presented is sufficiently clear to allow for an exact replication, and the authors have presented a fair consideration of sufficient outcome-neutral conditions. The analytic methods suggested are appropriate, and have been used in similar recent studies on screen time and wellbeing.

In general, I think that this will be a much-needed addition to the research literature on violent video game effects, a field that desperately needs preregistered studies so as to protect against undisclosed flexibility in analytic techniques (as the authors note in their introduction). We still

know very little about how violent game content has an impact, if any, on aggressive behaviour. As such, the present study has the potential to drive forward a new way of thinking about video games effects research.

By and large, the authors present a strong preregistration plan. I do think the study needs some additional tweaks, however.

1) Given that the study is explicitly looking at short-term variations in aggressive behaviour, it would be useful to have some sort of baseline measure of pre-existing aggression in the adolescent sample. I appreciate that this is difficult, given the cross-sectional nature of the design, but it is a limitation that certainly needs to be addressed in some way. I wonder whether use of something like the trait component of the state-trait anger expression inventory (STAXI) would help in this regard?

2) I'm torn on the video game violence measure. In a way, it is a very elegant design – game violence is usually poorly operationalised in these sorts of studies, and by using PEGI classifications as a basis, the authors are tying the violence categorisation to something that makes realistic sense in terms of parental game choices. On the other hand, I'm slightly worried that coding game violence as either 1 (violent) or 0 (not-violent) is too broad, and might have the potential to lose important information about specific game content that could be driving any associations with aggressive behaviour. For example, under the PEGI system, both World of Warcraft and Call of Duty are classed as 'violent' games. In reality, the way in which in-game violence plays out in these games is very different. I wonder, as a way to get over this issue while still retaining the elegance of using PEGI ratings, whether a measure of violence could be developed that takes into account the PEGI age rating? To go back to my example, World of Warcraft is PEGI 12, whereas Call of Duty is PEGI 18. In that sense then, they are categorised as different levels of violence.

As a simple tweak then, would it be appropriate to have 4 levels of the explanatory variable: 0 (no violence badge); 1 (violence badge, 12 rating); 2 (violence badge, 16 rating); 3 (violence badge, 18 rating)? This might also have the advantage of considering age-appropriateness as a potential factor; the proposed adolescent group will be aged 14-15, so in theory shouldn't be playing 16 or 18-rated games.

I look forward to further correspondence from the authors.

Signed,

Dr Pete Etchells

Reviewer: 4

Comments to the Author(s)

Dear Dr. Przybylski and Dr. Weinstein,

It is a pleasure to be able to review this project and to be given the opportunity to engage with you at this stage. My compliments on formulating a concise, cogent, and much-needed call for the study you are proposing. Studies of the scale and methodological rigor you present are much needed, and it is encouraging to see such competent researchers rise to the challenge.

I was left wondering a couple of things regarding your study:

1) The literature on violent video gaming has recently expanded into examining the influence of

cooperative and competitive gaming. Given the proposed size of your study's sample, I think you are in the unique position to be able to add clarity and great value by including these as potential mediating variables. Was this a consideration of yours? And if so, what influenced your decision not to investigate these variables? (Related to this, your study already asks participants whether the game they have been playing was being played in single or multiplayer mode.)

2) I commend your choice to rely on the PEGI ratings. I wonder whether you think it is important to distinguish between different types of violence ratings applied by PEGI (e.g. "cartoon violence" vs. "realistic violence").

I also have a recommendation for your methods:

Based on past studies conducted by myself and my colleagues, you may want to ask for more specificity when asking participants to nominate the games they played most over the past month. While devoted gamers are likely to be very specific about the games they played, more casual gamers are apt to list generic titles (e.g. "Mario") or to simply list the name of websites that host large numbers of simple Flash games and mini-games. You may therefore wish to give your participants a bit more directive with this item. I also miss the wording for how you will ask participants to describe whether they play single player or multiplayer (described as "variable 4" on page 5, but not listed in Appendix B).

Finally, some suggestions for very minor wording tweaks:

-Page 1, Paragraph 1. "Following preregistered analysis plan multiple regression analyses will test..." Add a comma after "plan".

-Page 4, Paragraph 3. "This survey-based study will... measure youth aggression with caregiver responses to a widely used in behavioural screening questionnaire..." I think this would read better with the word "questionnaire" moved to after "used".

Thank you for your work,
Best of luck,
Adam Lobel, PhD

Author's Response to Decision Letter for (RSOS-170903.R0)

See Appendix A.

RSOS-171259.R0

Review form: Reviewer 1 (Christopher Ferguson)

Is the language acceptable?

Yes

Do you have any ethical concerns with this paper?

No

Have you any concerns about statistical analyses in this paper?

Yes

Recommendation?

Accept with minor revision

Comments to the Author(s)

I think that the current proposal is much improved. I do have a few other comments however, that would need to be addressed before this is ready to go.

First, the current inclusion of the GAM theoretical model is problematic and uses biased language. For instance, the statement "Both reviews (24) and recent studies (25) using the GAM lens find consistent, though modest, support for the idea violent gaming is linked to human aggression. These relations are conceptually replicated in motivation research by studies indicating aggressive individuals find violent games compelling (26) and violent games, because they are complex can foment aggressive..." is simply incorrect. Both many recent review and individual studies (see recent studies by Grant Devilly, Mary Ballard, Aaron Drummond,...as well as Przybylski et al., 2014) that have explicitly failed to confirm the GAM. The idea that tests of the game find "consistent but modest" support for links with aggression is simply false. The authors need to be more honest about the problems with the GAM, the numerous null studies, and specific critiques of the GAM (e.g. Ferguson & Dyck, 2012).

Further, the authors also need to note that there are alternative theories regarding video game effects, including Self-Determination Theory, the Catalyst Model and Mood Management Theory which have differing views of the effects of games on aggression.

To be honest, I think this addition, as currently worded, was a serious backsliding of the quality of this proposal and *really* needs to be fixed.

Second, the authors really need to include gender as a covariate in their regression model, not conduct separate gender models. From my own experience working in this area, the issue of multicollinearity does not remotely arise when using both violent games and gender as predictors in a regression model.

Third, I understand that effect size cut-offs can be arbitrary, but the authors really do need to identify something, not just equivalence testing. There are several guidelines out there for the interpretation of trivial effects...they differ a bit, but I suspect the authors here could develop a kind of tiered system.

Despite my concerns in these areas, I remain very optimistic that the authors can easily address them and carry out an important study.

Signed,
Chris Ferguson

Review form: Reviewer 2**Is the language acceptable?**

Yes

Do you have any ethical concerns with this paper?

No

Have you any concerns about statistical analyses in this paper?

No

Recommendation?

Accept in principle

Comments to the Author(s)

I feel the authors responded to my previous comments in a satisfactory manner. Although I still have slight pause about labeling the outcome as a measure of "aggression" rather than "conduct problems," I concede that there is sufficient justification for using the conduct problems subscale of the SDQ. I strongly feel this proposed study will be an important contribution to this literature regardless of the results obtained.

Review form: Reviewer 3 (Peter Etchells)

Is the language acceptable?

Yes

Do you have any ethical concerns with this paper?

No

Have you any concerns about statistical analyses in this paper?

No

Recommendation?

Accept in principle

Comments to the Author(s)

Thank you for such a considered response to the reviewer comments. I think the inclusion of BPAS will present a useful control for baseline aggression. Further, the inclusion of PEGI ratings to assess total time spent playing violent games will, I think, add a fine-grained analysis to the study that is currently missing in the wider research literature. I look forward to reading the Stage 2 submission.

Signed,

Dr Pete Etchells

Decision letter (RSOS-171259.R0)

25-Sep-2017

Dear Dr Przybylski

On behalf of the Editors, I am pleased to inform you that your Manuscript RSOS-171259 entitled

"Violent Video Game Engagement and Adolescents' Aggressive Behaviour: A Registered Report" has been accepted in principle for publication in Royal Society Open Science subject to minor revision in accordance with the referee and editor suggestions. Please find their comments at the end of this email.

The reviewers and handling editors have recommended in principle acceptance, but also suggest some minor revisions to your manuscript. Therefore, I invite you to respond to the comments and revise your manuscript.

Please you submit the revised version of your manuscript within 7 days (i.e. by the 03-Oct-2017). If you do not think you will be able to meet this date please let me know immediately.

Full author guidelines can be found here
<http://rsos.royalsocietypublishing.org/content/registered-reports>.

Best wishes

Alice Power
Editorial Coordinator
Royal Society Open Science

on behalf of Chris Chambers
Subject Editor, Royal Society Open Science
openscience@royalsociety.org

Associate Editor Comments to Author:

The manuscript was returned to three of the original reviewers. Two are now satisfied with the submission and recommend IPA, while Reviewer 1 recommends some additional minor amendments to the theoretical framing and statistical analysis. Provided the next revision sufficiently addresses these remaining points, IPA should be forthcoming without requiring further in-depth review.

Reviewer comments to Author:

Reviewer: 2

Comments to the Author(s)

I feel the authors responded to my previous comments in a satisfactory manner. Although I still have slight pause about labeling the outcome as a measure of "aggression" rather than "conduct problems," I concede that there is sufficient justification for using the conduct problems subscale of the SDQ. I strongly feel this proposed study will be an important contribution to this literature regardless of the results obtained.

Reviewer: 1

Comments to the Author(s)

I think that the current proposal is much improved. I do have a few other comments however, that would need to be addressed before this is ready to go.

First, the current inclusion of the GAM theoretical model is problematic and uses biased language. For instance, the statement "Both reviews (24) and recent studies (25) using the GAM lens find consistent, though modest, support for the idea violent gaming is linked to human aggression. These relations are conceptually replicated in motivation research by studies indicating aggressive individuals find violent games compelling (26) and violent games, because they are complex can foment aggressive..." is simply incorrect. Both many recent review and individual studies (see recent studies by Grant Devilly, Mary Ballard, Aaron Drummond,...as well as Przybylski et al., 2014) that have explicitly failed to confirm the GAM. The idea that tests of the game find "consistent but modest" support for links with aggression is simply false. The authors need to be more honest about the problems with the GAM, the numerous null studies, and specific critiques of the GAM (e.g. Ferguson & Dyck, 2012).

Further, the authors also need to note that there are alternative theories regarding video game effects, including Self-Determination Theory, the Catalyst Model and Mood Management Theory which have differing views of the effects of games on aggression.

To be honest, I think this addition, as currently worded, was a serious backsliding of the quality of this proposal and *really* needs to be fixed.

Second, the authors really need to include gender as a covariate in their regression model, not conduct separate gender models. From my own experience working in this area, the issue of multicollinearity does not remotely arise when using both violent games and gender as predictors in a regression model.

Third, I understand that effect size cut-offs can be arbitrary, but the authors really do need to identify something, not just equivalence testing. There are several guidelines out there for the interpretation of trivial effects...they differ a bit, but I suspect the authors here could develop a kind of tiered system.

Despite my concerns in these areas, I remain very optimistic that the authors can easily address them and carry out an important study.

Signed,
Chris Ferguson

Reviewer: 3

Comments to the Author(s)

Thank you for such a considered response to the reviewer comments. I think the inclusion of BPAS will present a useful control for baseline aggression. Further, the inclusion of PEGI ratings to assess total time spent playing violent games will, I think, add a fine-grained analysis to the study that is currently missing in the wider research literature. I look forward to reading the Stage 2 submission.

Signed,
Dr Pete Etchells

Author's Response to Decision Letter for (RSOS-171259.R0)

Please see letter.

RSOS-171474.R0

Decision letter (RSOS-171474.R0)

28-Sep-2017

Dear Dr Przybylski

On behalf of the Editor, I am pleased to inform you that your Manuscript RSOS-171474 entitled "Violent Video Game Engagement and Adolescents' Aggressive Behaviour: A Registered Report" has been accepted in principle for publication in Royal Society Open Science.

You may now progress to Stage 2 and complete the study as approved. We would be grateful if you could now update the journal office as to the anticipated completion date of your study.

Following completion of your study, we invite you to resubmit your paper for peer review as a Stage 2 Registered Report. Please note that your manuscript can still be rejected for publication at Stage 2 if the Editors consider any of the following conditions to be met:

- The results were unable to test the authors' proposed hypotheses by failing to meet the approved outcome-neutral criteria
- The authors altered the Introduction, rationale, or hypotheses, as approved in the Stage 1 submission
- The authors failed to adhere closely to the registered experimental procedures
- Any post-hoc (unregistered) analyses were either unjustified, insufficiently caveated, or overly dominant in shaping the authors' conclusions
- The authors' conclusions were not justified given the data obtained

We encourage you to read the complete guidelines for authors concerning Stage 2 submissions at <http://rsos.royalsocietypublishing.org/content/registered-reports>. Please especially note the

requirements for data sharing and that withdrawing your manuscript will result in publication of a Withdrawn Registration.

Once again, thank you for submitting your manuscript to Royal Society Open Science and I look forward to receiving your stage 2 submission. If you have any questions at all, please do not hesitate to get in touch. We look forward to hearing from you shortly with the anticipated submission date for your stage two manuscript.

Kind regards,

Alice Power
Editorial Coordinator
Royal Society Open Science

on behalf of Chris Chambers
Registered Reports Editor, Royal Society Open Science
openscience@royalsociety.org

Author's Response to Decision Letter for (RSOS-171474.R0)

See Appendix B.

RSOS-171474.R1 (Revision)

Review form: Reviewer 1 (Christopher Ferguson)

Is the language acceptable?

Yes

Do you have any ethical concerns with this paper?

No

Have you any concerns about statistical analyses in this paper?

No

Recommendation?

Accept as is

Comments to the Author(s)

The current study is the completion of a registered research report looking at violent game playing and aggression in youth. The study appears to have followed the registration, managed to obtain an impressive sample and has been conducted in a thorough and convincing manner. I am sure that this study will have a major impact on debates about violent game impacts. I have just a few very minor observations:

The authors mention the APA Task Force Report from 2015, but they might also want to note that the APA's own media psychology division in 2017 released their own policy statement implicitly rebuking the APA as a whole and recommending against linking violent games to serious aggression in real life.

The authors do a nice job summarizing a complex research field as well as important methodological issues for this field. The only place they make an error is in concluding (as they repeat in the discussion) that there is only 1 previous preregistered study of VVG effects. The McCarthy study is certainly a worthwhile one, but it's not the only preregistered study. I've done several myself dating back to 2015, and I believe James Ivory has at least one.

It's good that the authors note some of the problems with methodological flexibility in prior research. The example of the Singapore dataset is a striking one.

The statistical analyses are all well done. Their method of calculating VVG exposure is clearly cutting edge and they are right not to use participant reports.

As a minor issue Page 10, p-values for ESB ratings may be swapped for the linear and parabolic outcomes.

Other than that, this is ready to go. I congratulate the authors on excellent work!

Signed,
Chris Ferguson

Review form: Reviewer 2

Is the language acceptable?

Yes

Do you have any ethical concerns with this paper?

No

Have you any concerns about statistical analyses in this paper?

No

Recommendation?

Accept as is

Comments to the Author(s)

This Stage 2 manuscript is written clearly, appeared to followed the agreed-upon methods and analyses, and has properly interpreted the results. I believe the authors have done an excellent job and have no issues with the manuscript being published as-is.

I have three minor points that the authors could consider. First, as an additional positive control, I would note that Trait Physical Aggression was positively associated with participants' aggressive behavior, $r = .62$, $p < .001$. This adds credence to the notion that participants were responding sensibly and that the measure of aggressive behavior is valid. Second, it seems buried in the manuscript that aggressive behaviors are positively associated with violent game engagement ($r = .08$), overall game engagement ($r = .11$), and subjective game engagement ($r = .25$). I have no doubt that these numbers will be seized upon by proponents of the "violent video game-

aggression effect" to argue that your conclusions are misguided. Finally, I am a bit ambivalent about the paragraph in arguing for adopting $d = +0.50$ as the benchmark for a meaningful effect. I can easily see effects being smaller than that being meaningful, especially if that effect is describing an aggressive behavior. I would consider just taking that paragraph out of the manuscript.

Whether the data are able to test the authors' proposed hypotheses by passing the approved outcome-neutral criteria (such as absence of floor and ceiling effects or success of positive controls)

See comment above

Whether the Introduction, rationale and stated hypotheses are the same as the approved Stage 1 submission

Yes, these aspects appear the same as the Stage 1 submission.

Whether the authors adhered precisely to the registered experimental procedures

Yes, the authors adhered to these procedures.

Where applicable, whether any unregistered exploratory statistical analyses are justified, methodologically sound, and informative

Yes. I feel that the addition of the prosocial behaviors outcome was an especially informative addition.

Whether the authors' conclusions are justified given the data

Yes.

Please note that editorial decisions will not be based on the perceived importance, novelty, or clarity of the results.

Review form: Reviewer 3 (Peter Etchells)

Is the language acceptable?

Yes

Do you have any ethical concerns with this paper?

No

Have you any concerns about statistical analyses in this paper?

No

Recommendation?

Accept as is

Comments to the Author(s)

The authors present a well-powered study that has been conducted and analysed in accordance with the procedures initially outlined in their Stage 1 submission, and pass the outcome-neutral conditions. I note that a number of minor changes to the text of the introduction have been made, but these are restricted to synonyms and clarifications, and do not alter the substance of this section in any way. No changes were made to the stated hypotheses, and the rationale remains that same as that outlined at Stage 1.

As far as I can ascertain, the authors clearly and strictly adhered to their registered experimental methodology and procedures. They were unable to calculate the inflection points for violent gaming effects, as the initial parabolic term was non-significant, making this sensitivity analysis redundant. This is fine, and is in line with their Stage 1 outline.

The authors conducted two additional exploratory analyses which were not registered in the Stage 1 submission; these are clearly denoted as such at Stage 2. Both analyses – that is, using US-based ratings (ESRB) as opposed to the European-based (PEGI) ratings used in the pre-registered analysis, and assessing prosocial behaviour (as opposed to aggressive behaviour) as an outcome measure – make sense within the context of the theoretical rationale, and add a further layer of robustness to their analyses by (a) making the findings more internationally-relevant, and (b) assessing for positive vs. negative effects of video game play. In this regard, I am satisfied that the additional analyses are methodologically sound, and are clearly relevant.

The authors also provide a clear and objective overview of their results in the discussion section. The conclusions that they draw – that adolescent ‘violent’ video game play has no statistically-significant (or practically significant) effects on either aggressive or prosocial behaviour – is clearly borne out by the data that they have presented, and they present a well-reasoned and evidence-backed summary of the study. As I noted in my Stage 1 reviews, this study presents a much-needed addition to an area of research that, in general, would benefit hugely from further preregistered analyses (and RR submissions). Aside from the results themselves adding significant value to our understanding of the topic, the fact that the authors have demonstrated that it is clearly achievable to conduct a Registered Report concerning video games effects will hopefully provide an impetus for other researchers to hold themselves to similarly high standards in the future.

Decision letter (RSOS-171474.R1)

03-Jan-2019

Dear Dr Przybylski:

On behalf of the Editor, I am pleased to inform you that your Stage 2 Registered Report RSOS-171474.R1 entitled "Violent Video Game Engagement is not Associated with Adolescents' Aggressive Behaviour: Evidence from a Registered Report" has been deemed suitable for publication in Royal Society Open Science subject to minor revision in accordance with the referee suggestions. Please find the referees' comments at the end of this email.

The reviewers and Subject Editor have recommended publication, but also suggest some minor revisions to your manuscript. Therefore, I invite you to respond to the comments and revise your manuscript.

Please also ensure that all the below editorial sections are included where appropriate -- if any section is not applicable to your manuscript, please can we ask you to nevertheless include the heading, but explicitly state that the heading is inapplicable. An example of these sections is attached with this email.

- Ethics statement

- Data accessibility

It is a condition of publication that all supporting data are made available either as supplementary information or preferably in a suitable permanent repository. The data

accessibility section should state where the article's supporting data can be accessed. This section should also include details, where possible of where to access other relevant research materials such as statistical tools, protocols, software etc can be accessed. If the data has been deposited in an external repository this section should list the database, accession number and link to the DOI for all data from the article that has been made publicly available. Data sets that have been deposited in an external repository and have a DOI should also be appropriately cited in the manuscript and included in the reference list.

If you wish to submit your supporting data or code to Dryad (<http://datadryad.org/>), or modify your current submission to dryad, please use the following link:
[http://datadryad.org/submit?journalID=RSOS&manu=\(Document not available\)](http://datadryad.org/submit?journalID=RSOS&manu=(Document not available))

- **Competing interests**

- **Authors' contributions**

- **Acknowledgements**

- **Funding statement**

Because the schedule for publication is very tight, it is a condition of publication that you submit the revised version of your manuscript within 7 days (i.e. by the 11-Jan-2019). If you do not think you will be able to meet this date please let me know immediately.

When submitting your revised manuscript, you will be able to respond to the comments made by the referees and upload a file "Response to Referees" in "Section 6 - File Upload". You can use this to document any changes you make to the original manuscript. In order to expedite the

processing of the revised manuscript, please be as specific as possible in your response to the referees.

Please note that Royal Society Open Science will introduce article processing charges for all new submissions received from 1 January 2018. Registered Reports submitted and accepted after this date will ONLY be subject to a charge if they subsequently progress to and are accepted as Stage 2 Registered Reports. If your manuscript is submitted and accepted for publication after 1 January 2018 (i.e. as a full Stage 2 Registered Report), you will be asked to pay the article processing charge, unless you request a waiver and this is approved by Royal Society Publishing. You can find out more about the charges at <http://rsos.royalsocietypublishing.org/page/charges>. Should you have any queries, please contact openscience@royalsociety.org.

on behalf of Professor Chris Chambers (Registered Reports Editor, Royal Society Open Science)
openscience@royalsociety.org

Associate Editor Comments to Author (Professor Chris Chambers):

Associate Editor: 1

Comments to the Author:

The Stage 2 manuscript was returned to three of the expert reviewers who assessed the Stage 1 protocol. All are very positive about the submission and judge that the Stage 2 criteria are largely met. However, the reviewers offer some useful final suggestions to consider in the interpretation and conclusions (e.g. the point about considering $d=0.5$ as a meaningful cutoff) as well as some other minor corrections and clarifications. A minor revision is therefore recommended. Provided the revision addresses the reviewers' points, final Stage 2 acceptance should be forthcoming without requiring further in-depth review.

Comments to Author:

Reviewer: 2

Comments to the Author(s)

This Stage 2 manuscript is written clearly, appeared to follow the agreed-upon methods and analyses, and has properly interpreted the results. I believe the authors have done an excellent job and have no issues with the manuscript being published as-is.

I have three minor points that the authors could consider. First, as an additional positive control, I would note that Trait Physical Aggression was positively associated with participants' aggressive behavior, $r = .62$, $p < .001$. This adds credence to the notion that participants were responding sensibly and that the measure of aggressive behavior is valid. Second, it seems buried in the manuscript that aggressive behaviors are positively associated with violent game engagement ($r = .08$), overall game engagement ($r = .11$), and subjective game engagement ($r = .25$). I have no doubt that these numbers will be seized upon by proponents of the "violent video game-aggression effect" to argue that your conclusions are misguided. Finally, I am a bit ambivalent about the paragraph in arguing for adopting $d = +0.50$ as the benchmark for a meaningful effect. I can easily see effects being smaller than that being meaningful, especially if that effect is describing an aggressive behavior. I would consider just taking that paragraph out of the manuscript.

Whether the data are able to test the authors' proposed hypotheses by passing the approved outcome-neutral criteria (such as absence of floor and ceiling effects or success of positive controls)

See comment above

Whether the Introduction, rationale and stated hypotheses are the same as the approved Stage 1 submission

Yes, these aspects appear the same as the Stage 1 submission.

Whether the authors adhered precisely to the registered experimental procedures

Yes, the authors adhered to these procedures.

Where applicable, whether any unregistered exploratory statistical analyses are justified, methodologically sound, and informative

Yes. I feel that the addition of the prosocial behaviors outcome was an especially informative addition.

Whether the authors' conclusions are justified given the data

Yes.

Please note that editorial decisions will not be based on the perceived importance, novelty, or clarity of the results.

Reviewer: 1

Comments to the Author(s)

The current study is the completion of a registered research report looking at violent game playing and aggression in youth. The study appears to have followed the registration, managed to obtain an impressive sample and has been conducted in a thorough and convincing manner. I am sure that this study will have a major impact on debates about violent game impacts. I have just a few very minor observations:

The authors mention the APA Task Force Report from 2015, but they might also want to note that the APA's own media psychology division in 2017 released their own policy statement implicitly rebuking the APA as a whole and recommending against linking violent games to serious aggression in real life.

The authors do a nice job summarizing a complex research field as well as important methodological issues for this field. The only place they make an error is in concluding (as they repeat in the discussion) that there is only 1 previous preregistered study of VVG effects. The McCarthy study is certainly a worthwhile one, but it's not the only preregistered study. I've done several myself dating back to 2015, and I believe James Ivory has at least one.

It's good that the authors note some of the problems with methodological flexibility in prior research. The example of the Singapore dataset is a striking one.

The statistical analyses are all well done. Their method of calculating VVG exposure is clearly cutting edge and they are right not to use participant reports.

As a minor issue Page 10, p-values for ESB ratings may be swapped for the linear and parabolic outcomes.

Other than that, this is ready to go. I congratulate the authors on excellent work!

Signed,
Chris Ferguson

Reviewer: 3

Comments to the Author(s)

The authors present a well-powered study that has been conducted and analysed in accordance with the procedures initially outlined in their Stage 1 submission, and pass the outcome-neutral conditions. I note that a number of minor changes to the text of the introduction have been made, but these are restricted to synonyms and clarifications, and do not alter the substance of this section in any way. No changes were made to the stated hypotheses, and the rationale remains that same as that outlined at Stage 1.

As far as I can ascertain, the authors clearly and strictly adhered to their registered experimental methodology and procedures. They were unable to calculate the inflection points for violent gaming effects, as the initial parabolic term was non-significant, making this sensitivity analysis redundant. This is fine, and is in line with their Stage 1 outline.

The authors conducted two additional exploratory analyses which were not registered in the Stage 1 submission; these are clearly denoted as such at Stage 2. Both analyses – that is, using US-based ratings (ESRB) as opposed to the European-based (PEGI) ratings used in the pre-registered

analysis, and assessing prosocial behaviour (as opposed to aggressive behaviour) as an outcome measure – make sense within the context of the theoretical rationale, and add a further layer of robustness to their analyses by (a) making the findings more internationally-relevant, and (b) assessing for positive vs. negative effects of video game play. In this regard, I am satisfied that the additional analyses are methodologically sound, and are clearly relevant.

The authors also provide a clear and objective overview of their results in the discussion section. The conclusions that they draw – that adolescent ‘violent’ video game play has no statistically-significant (or practically significant) effects on either aggressive or prosocial behaviour – is clearly borne out by the data that they have presented, and they present a well-reasoned and evidence-backed summary of the study. As I noted in my Stage 1 reviews, this study presents a much-needed addition to an area of research that, in general, would benefit hugely from further preregistered analyses (and RR submissions). Aside from the results themselves adding significant value to our understanding of the topic, the fact that the authors have demonstrated that it is clearly achievable to conduct a Registered Report concerning video games effects will hopefully provide an impetus for other researchers to hold themselves to similarly high standards in the future.

Author's Response to Decision Letter for (RSOS-171474.R1)

See Appendix C.

Decision letter (RSOS-171474.R2)

18-Jan-2019

Dear Dr Przybylski:

It is a pleasure to fully accept your Stage 2 Registered Report entitled "Violent Video Game Engagement is not Associated with Adolescents' Aggressive Behaviour: Evidence from a Registered Report" in its current form for publication in Royal Society Open Science.

on behalf of Professor Chris Chambers (Subject Editor)
openscience@royalsociety.org

Appendix A

February 4, 2019

Dear Alice Power and the editorial team,

Thank you very much for your thorough and helpful review of our manuscript entitled: "Violent Video Game Engagement and Adolescents' Aggressive Behaviour: A Registered Report"(RSOS-170903), as well as your invitation to revise it. We have carefully taken into consideration your comments, as well as those of the four experts who also reviewed the paper. With these inputs, we have made revisions and additions to our study's introduction, measures, and analytic strategy. We think these changes have improved the program of research. Below we address your specific concerns, and also give responses and actions with respect to each of them. We begin with your points, and then review the comments of the four expert reviewers.

Editor Comments

"Four expert reviewers have now assessed the manuscript. All are positive but offer a range of critical suggestions that cover the full spectrum of Stage 1 review criteria. Two issues in particular have been highlighted by the editorial team. First, as noted by Reviewer 1, the authors should consider the inclusion of control variables. Second, Reviewer 2 makes the important point that the outcome measure does not provide a very precise measure of aggression, and indeed mostly captures non-aggressive conduct. Please be sure to address these two major points in addition to the full range of detailed comments raised by all reviewers."

Point 1

Three of the reviewers, Prof. Ferguson, Reviewer 2, and Dr. Etchells raised the issue of using an individual differences measure of aggressive personality as a covariate in our hypothesis testing models. A review of the relevant literatures indicated there were a range of possible measures to assess this factor, i.e. trait-level, measures of aggressive personality. We ruled out many of these because there were commercial/licensed scales which do not make their computational method public, and would be insufficiently transparent for the RR format. We have identified one good candidate, a brief version of the Buss-Perry aggression scale (1), which taps into physical aggression, verbal aggression, hostility, and angry emotions. We have added this measure to our study and to our hypothesis testing.

Point 2

Reviewer 2 raised an interesting point concerning our measure of behavioural aggression. He or she suggested we supplement our criterion variable, the conduct problems subscale of the Strengths of Difficulties Questionnaire (2), with a measure of interpersonal physical aggression. We have considered this point carefully and decided the SDQ was the best choice to measure the construct at this stage of research for three reasons. We detail our rationale in our response to Reviewer 2 (Point 5) below.

Reviewer: 1

Overall comment

“I enjoyed reading this preregistered replication proposal for RSOS. The proposal is well written, covers the prior literature accurately, and makes a good case for their study. The authors should be congratulated for using preregistered designs in their analyses. I do, however, have a few comments that might improve their design.”

We thank Prof. Ferguson for his generally positive take on our work.

Point 1

“First, as a minor note, on page 2 the authors conclude that Hilgard et al., report an effect size for video games on aggression as $r = .21$. I don't think that's exactly correct, looking at Table 3 from the Hilgard paper. From Hilgard, adjusted effect size estimates vary depending upon correction method used, so I'm not sure you can point to a single adjustment approach. I'm guessing that the authors here may be referring to the “full” studies for aggressive behavior in correlational methods which is indeed $.21$ (but confidence interval is $.12, .28$) using PET. You might just want to be clearer exactly where that effect size is coming from as this confused me a bit. Effect sizes were much lower for experimental studies, but that may not be relevant here for a correlational study.”

The estimate we are using for the power analysis for our proposed study is based on the naïve effect-size estimates in Table 1 (3) for cross sectional studies relating violent game play to self-report measures of aggressive ($k = 37, n = 29,113$) which yields an estimated an effect size of $r = .21$ [95 CI = $.20, .22$]. We avoided using the adjusted effect size estimates, (Table 3) because the estimated effect size was the same ($r = .21$) but the precision of the estimate was greatly reduced: The confidence interval was nearly 8x wider [95 CI = $.12, .28$]. We have revised the manuscript to make this clear (p.2).

Point 2

*“As to the design, I had a few comments. First, I would *strongly* urge the authors to include control variables identified as important in the literature. This includes not just gender (which is, indeed, important), but also mental health (particularly depression), personality (trait aggression), family environment (psychological abusiveness between parents has been observed as important in prior studies), and peer delinquency. This could be achieved with just a few additional variables. At present, the current design’s main weakness is the third-variable problem. The current design is too basic. But I think this will be easy for the authors to fix.”*

This is a very important point. Though we are limited on survey length, we believe adding the 12-item Buss-Perry Aggression Questionnaire (1), completed by adolescent respondents, will address allow us to quantify, and hold constant the variability Prof. Ferguson is highlighting, e.g. (4). We selected this measure over others because it is a widely used open source measure. In line with this, we will include the individual differences (i.e. trait-level), physical aggression, verbal aggression, anger, and hostility, as a control variables in our hypothesis-testing models.

Point 3

“Also, instead of separate regression models for boys and girls, I would simply include gender as a dummy coded variable in the regression model.”

We understand the root of this suggestion. Based on the existing literature, we are concerned that gender will be highly collinear with violent game preference, play behaviour, and the outcome of interest. Because we have sufficient statistical power at our present sample size, we would prefer to keep the statistical models as simple as possible to minimise the chance of introducing statistical artefacts which will impact the false discovery rates or suppression effects (5). We can of course include such models, using gender as a dummy-coded variable, as an exploratory analysis as too can researchers building on the data we generate.

Point 4

“Third, I know this would eat up more time, but the current study is also “demand characteristic” ...although I grant having parents and kids fill out different forms helps some. Assuming that the parents and kids are both online taking the survey semi-together, the potential for DC remains high though. I’d suggest including some distractor questions to make the

hypotheses of the study less obvious, particularly to parents who may overrespond to aggression once they see their kids responding to video game questionnaires. I'd ask the kids about other hobbies, plus some other things, and ask the parents some irrelevant stuff about their kids too."

This is a good point. The questionnaires for this study will be interspersed with a range of questions about their experiences online and how their parents make rules about their use of smartphones, tablets, and social media platforms. Because the primary predictive factor, game content, will be coded by the researcher it should be difficult for participants to guess the purpose of the study.

Point 5

"Lastly, I encourage the authors to predetermine effect size cut-offs for practical significance. Particularly with such a large sample, there's the potential for getting "statistically significant" results that are nonetheless trivial...or due to some of the noise in the study (like demand characteristics!)...not real effects. An $r = .10$ would be a bare minimum cut-off for interpretation, with $.20$ probably being better if you want to talk about clinically significant effects."

We agree that this is a concern. In line with this we will conduct equivalence testing (6) comparing the effect sizes we observe to the best existing estimate of present in the literature, $r = .21$ (95% CI = $.20$ to $.22$). This should let us know if the effects we observe are the equivalent, superior, or inferior to what meta-analytic evidence suggests (p.7). We are open to examining additional equivalence tests could be conducted on an exploratory basis. We have searched the literature for other empirical benchmarks but have not found promising candidates to base these r and 95% CI estimates on.

Closing thought

"I hope these comments are helpful. I do think this would be an important study, particularly if we get a few more control variables in there! Signed, Chris Ferguson, Stetson University."

We thank Prof. Ferguson for his helpful comments and for signing his review.

Overall Comment

“I have now had an opportunity to review the proposed study “Violent video game engagement and adolescents’ aggressive behavior: A registered report” for possible publication in Royal Society Open Science. Consistent with RSOS reviewing guidelines, I have addressed the following items in my review. The significance of the proposed research question is undoubtedly important. The relationship between exposure to violent media (with violent video games being one specific type of media) and aggression has theoretical and societal relevance. And, unfortunately, the extant research into this effect has left researchers with an uncertain empirical foundation to build upon (indeed, some people believe this empirical foundation is strong and some people believe it is a complete disaster). This has left many individuals with the impression that we need to (re?)establish even the most basic effects relevant to this hypothesis: Is there even a bi-variate association between playing violent video games and aggression? In my opinion, this means that we need pre-registered and “open” studies such as this one even if it seems to rehash effects that some individuals believe are well-established.”

We thank Reviewer 2 for his or her appreciation for the need for our study in light of the existing research base.

Point 1

“In making the case for the significance of the research question, the authors should cite the recent article by Anderson et al. (2017). Anderson, C. A., Suzuki, K., Swing, E. L., Groves, C. L., Gentile, D. A., Prot, S., ... & Jelic, M. (2017). Media Violence and Other Aggression Risk Factors in Seven Nations. Personality and Social Psychology Bulletin, 43(7), 986-998.”

This is an interesting paper that we missed, we have included it in our revised manuscript.

Point 2

“The logic and rationale of the proposed hypotheses are not well articulated. I believe that a lot more could be done to argue why it is plausible that exposure to violent video games could cause subsequent aggression. The current Introduction discusses relevant meta-analyses and general criticisms of this area of research; however, there is little discussion about the theoretical underpinnings of the stated hypothesis. This would probably mean discussing hypotheses that are derived from social-cognitive models of

aggression (e.g., exposure to violent video games reinforces aggressive responding/desensitizes participants to violence, etc.). Also, although I am sure the authors intended to be skeptical and critical, the Introduction seems to come off as a bit biased against the existence of the effect. I really dislike making vague statements about authors' writing style, but there does seem to be a subtle negative tone that seeps into the Introduction."

This was an omission. We agree that the introduction could have introduced a clearer theoretical basis for the hypothesis including the social cognitive models, namely the dominant perspective the general aggression model (p.2). The revised version of the paper now provides a brief overview of the GAM and literature supporting the view that violent gaming could be positively and meaningfully related to human aggression.

Point 3

*"I can imagine that some people may quibble with the fact that the primary hypothesis is *merely* a correlation and believe that *more sophisticated* moderators of this believed-to-be-well-established effect should be tested, but I believe the simplicity of the hypothesis is also the strength of the hypotheses. Although it is true that moderators may strengthen or weaken the relationship between violent video game exposure and aggression, the cited meta-analyses clearly argue the simple bi-variate relationship between violent video game exposure and aggression would be expected."*

We agree and thank the reviewer for raising this point.

Point 4

*"My only comment about the conceptual hypotheses is that the authors imply, but do not explicitly state, the direction of the expected relationship. The authors should explicitly state they expect a *positive* relationship between violent video game exposure and aggression. Also, the authors should specify the expected direction of the parabolic relationship (i.e., does the relationship strengthen or weaken as violent video game increases?). I could post-hoc spin a plausible sounding explanation for either direction of a parabolic relationship (if it emerged)."*

The revised version of the manuscript now makes it explicit that we expect a positive relation between violent video game play and the study outcome and an 'open up' parabolic relation to the study outcome (p.4).

Point 5

*“My major criticism is not in the hypotheses per se, but in the operationalization of the hypotheses. The outcome measure is the *Conduct Problems* subscale of the SDQ. To my read, there is only one item that actually measures aggression (i.e., Often fights with other children or bullies them). The other items seem to ask about non-aggressive conduct problems (e.g., often has temper tantrums or hot tempers; often lies or cheats; steals from home, school, or elsewhere). This is the same shortcoming as the Anderson et al. (2017) paper mentioned above that also claimed to measure aggressive behaviors. However, these items do NOT measure aggressive behaviors and, thus, are inappropriate for testing the hypotheses. This is a big, big deal. Is it too late to include parent reports of their child’s aggressive behaviors? Or is it possible that the authors can modify their hypotheses to state they are interested in conduct problems in general, not aggression specifically?”*

A minor point is the framing of the timeframe of the items. The child respondents will be self-reporting on their video game play in the past month and there does not seem to be a timeframe for parents’ reports of their children’s behaviors. Is it possible to tell parents to think about their child’s behavior in the past month? Or justify why you would not want to instruct the children and parents to think of the same timeframe when completing the study.”

This is a really interesting point. At its core, Reviewer 2 is correct in noting that only one of the five items comprising the conduct problems subscale of the Strengths and Difficulties Questionnaire (2) measures interpersonal physically aggressive behaviour (item 12). Though our methodological decision is not perfect, we believe the choice of this assessment strategy is the best one given the state of the existing video game and child aggression literatures for three reasons.

First, the conduct problems subscale is a measure of behavioral aggression was created to tap into a wide range of aggressive behaviours: Explosive aggressive outbursts (item 5), an (in)ability to self-regulate emotions (item 7), physical interpersonal aggression (item 12), and verbal and goal-directed interpersonal aggression (items 18 and 22). We believe it is important to keep the measure of aggressive behaviour in a general form because the effect size estimate we base our power analysis on uses a similar, though not validated, self-report measures of behavioural aggression, not explicitly interpersonal physical aggression. Second, the conduct problems subscale has been validated and used as a measure of behavioural aggression in dozens of studies in school and clinical settings. Given it’s validated in self-report form, as well as from teacher, parent, psychologist reports we want to avoid

modifying the measure with additional items because we want to preserve maximise generalisability to these studies. Third, nearly all of the alternative measures for assessing aggressive behaviour are either very long, e.g. the Development and Well-Being Assessment (DAWBA; 68-items), or they are closed-source, e.g. Children's Aggression Scale (CAS; Halperin, McKay). We were unable to find a suitable well-operationalised alternative measure that could be completed by parents and would be open. Because of this, the results using a proprietary scale would not be computationally reproducible without breaking copyright. As far as we are aware, there are no alternative suitable validated clinical measures. For these reasons we have a strong preference to using the SDQ. We agree that it is important for readers to know that we did not use a measure expressly focused on physical forms interpersonal behavioural aggression and we will note this as a limitation in the discussion.

We apologise for the confusion about the length of time parents were asked about. The version of the measures on the OSF was not the latest (the version in the appendices was), the timeframe for parent reports is for the past month (Appendix A).

Point 6

“This is a fairly straightforward design so I do not have any problems with the methods. The only aspect that gives me slight pause is that the methods rely heavily on the ability of the research firm to recruit participants. Data collection could be difficult because the unit of analysis is actually a child-parent dyad, which means that you could have issues with some dyads missing data from one person (see my comment below about pre-registering a plan to handle cases where one person in the dyad does not provide a response). Also, is there anything in place to ensure that respondents do not influence each other? For example, children and parents who complete the survey independently will likely give different responses than children and parents who sit down and complete the survey jointly. What instructions will children/parents be given when they are asked to participate? Will they be told to keep their responses confidential?”

Caregiver respondents will complete surveys before the adolescent participants. Participants will be asked to leave the room when the other member of the dyad is completing measures. This information has been added to the Materials and Methods section (p.5). Only data from complete dyads will be included as demographic quotas are filled during sampling.

Point 7

“Can the authors add more detail about how participants and their parents will be sampled? How will the research firm find and recruit participants? How will participants be compensated? Also, awesome that this will be a representative sample. Good job!!:”

In line with this point, we have added data to the study sampling plan in the Materials and Methods section (pp.4-5).

Point 8

“I would like to see information about the following aspects of the analysis pipeline. First, are there any exclusion criteria that will be considered. For example, if participants provide missing data, will the authors omit those participants or will the authors attempt to impute the missing values? If the latter, how will the authors impute data?”

In cases of missing data, participants will not be included in analyses on a case-wise basis.

Point 9

“Also, is the relationship between children and parents one-to-one? For example, is it possible to have more than one child respondent who share the same parent? How will the authors handle cases of children nested within parents? Randomly select one child? Take the oldest/youngest child? Analyze the data using multi-level modeling?”

Yes the relationship is one-to-one. Though it is indeed possible that a caregiver will have two adolescents meeting the age criteria (e.g. twins) the surveys will only be completed by dyads. In cases where two adolescents fit one will be asked one adolescent-parent pair will be polled.

Point 10

“Similarly, what happens if only a child or only a parent responds (effectively making some units of analysis to have missing predictor variables or missing outcome variables)? Will those participants be completely omitted from analyses?”

We believe the research firm will continue to collect data until the complete sample, consisting of caregiver-adolescent pairs is recruited.

Point 11

“Second, please have the authors justify why they will conduct two parallel analyses separately for males and females. From my perspective, it would seem that you would merely enter the child gender and the interaction term as a moderator in one common analysis. If the authors ultimately decide to conduct one common analysis with gender as a moderator, the authors also should specify how they will code for child gender (e.g., dummy coding where 0=female and 1=male, etc.).”

In line with our reply to Prof. Ferguson, Point 3, we anticipate that gender and aggressive behaviour (2), gender and violent game preference (7,8) will be highly correlated. In cases where researchers know measures and their relations will be collinear it is preferable to conduct parallel analyses instead of relying on statistical controls to correct estimates (5).

Point 12

Third, is there any other measures from the data collection process that could be used as “control” variables? For example, because this is a correlational method, it is plausible that there is a third-variable problem where an unmeasured variable (e.g., trait aggression) causes both exposure to violent video games and aggression. Now would be a good time to add those variables (if any) into planned analyses.

We agree with this point, also raised by Prof. Ferguson (point 2). We have now included a individual differences measure aggression based on adolescent self-report.

Point 13

“Is the exploratory analysis necessary? It seems to be merely a different way to test the primary hypothesis.”

We can omit this from the stage 1 submission if the editor agrees and conduct these analyses on an entirely exploratory basis.

Point 14

Have authors considered sufficient outcome-neutral conditions to test the stated hypotheses?

Is it possible to add a “positive control” or some instructional manipulation checks? For example, can the authors add an item like “I spend a lot of time playing video games” and see if this item positively correlates with their self-

reported amount of time playing video games. Although it is possible to have a divergence between objective measures of time spent in an activity (e.g., hours per week) and a subjective measure (e.g., I *feel* like I spend a lot of time doing an activity), I would put a high prior on observing a positive relationship between the two items. Something like this would demonstrate that the researchers are able to detect predictable relationships between variables with their data collection procedure, which would help interpret results of the tests of the primary hypotheses.

We agree, and we have added a check question to this effect (p.6).

Point 15

“In the Introduction the authors discuss the possibility of “mischievous responding”, but also don’t seem to address this problem beyond avoiding child self-reports. There is still issues about participants’ (both parents and children) attending to the questions, responding honestly, etc. that would seem to be important to consider here. If anything is added to potentially exclude participants who give bad data, there would need to be decisions made about how those exclusion criteria would be enforced (e.g., are dyads excluded if either the child or the parent provides bad data?).”

Reviewer 2 is correct that our approach addresses mischievous responding by splitting the predictive and criterion variables across adolescent and caregiver respondents. Though we are limited by interview time with the participants, we will add that attention checks and social desirability can be examined in future studies.

Closing thoughts

“In summary, assuming the authors would be able to add/modify the outcome variable to measure actual aggressive behaviors, I strongly support this project moving forward. These types of projects are critical to advancing our understanding of the effects of violent video game exposure and aggression.”

We thank reviewer 2 for his or her helpful comments and raising important points in this review.

Overall Comments

“The authors present a proposal for a well-powered study assessing whether there is any association between violent content in video games and recent aggressive behaviour in adolescents. The rationale for the study is sound, and the study would present a substantial advance in the field. The study as presented is sufficiently clear to allow for an exact replication, and the authors have presented a fair consideration of sufficient outcome-neutral conditions. The analytic methods suggested are appropriate, and have been used in similar recent studies on screen time and wellbeing. In general, I think that this will be a much-needed addition to the research literature on violent video game effects, a field that desperately needs preregistered studies so as to protect against undisclosed flexibility in analytic techniques (as the authors note in their introduction). We still know very little about how violent game content has an impact, if any, on aggressive behaviour. As such, the present study has the potential to drive forward a new way of thinking about video games effects research. By and large, the authors present a strong preregistration plan. I do think the study needs some additional tweaks, however.”

We thank Dr. Etchells for his largely positive take and attend to his points below.

Point 1

“Given that the study is explicitly looking at short-term variations in aggressive behaviour, it would be useful to have some sort of baseline measure of pre-existing aggression in the adolescent sample. I appreciate that this is difficult, given the cross-sectional nature of the design, but it is a limitation that certainly needs to be addressed in some way. I wonder whether use of something like the trait component of the state-trait anger expression inventory (STAXI) would help in this regard?”

In line with the points raised by Prof. Ferguson (point 2) and Reviewer 2 (point 12) we no include an individual differences measure of trait-level aggression and will include the measure as a covariate for our hypothesis testing.

Point 2

“I’m torn on the video game violence measure. In a way, it is a very elegant design – game violence is usually poorly operationalised in these sorts of

studies, and by using PEGI classifications as a basis, the authors are tying the violence categorisation to something that makes realistic sense in terms of parental game choices. On the other hand, I'm slightly worried that coding game violence as either 1 (violent) or 0 (not-violent) is too broad, and might have the potential to lose important information about specific game content that could be driving any associations with aggressive behaviour. For example, under the PEGI system, both World of Warcraft and Call of Duty are classed as 'violent' games. In reality, the way in which in-game violence plays out in these games is very different. I wonder, as a way to get over this issue while still retaining the elegance of using PEGI ratings, whether a measure of violence could be developed that takes into account the PEGI age rating? To go back to my example, World of Warcraft is PEGI 12, whereas Call of Duty is PEGI 18. In that sense then, they are categorised as different levels of violence. As a simple tweak then, would it be appropriate to have 4 levels of the explanatory variable: 0 (no violence badge); 1 (violence badge, 12 rating); 2 (violence badge, 16 rating); 3 (violence badge, 18 rating)? This might also have the advantage of considering age-appropriateness as a potential factor; the proposed adolescent group will be aged 14-15, so in theory shouldn't be playing 16 or 18-rated games."

We agree that the coding system we originally proposed might miss out on meaningful variance with respect to what gaming violence is in different games. We've updated our analysis plan including this proposal as an additional sensitivity analysis using this approach.

Closing thoughts

"I look forward to further correspondence from the authors. Signed, Dr Pete Etchells"

We thank Dr. Etchells for his valuable input and for signing his review.

Overall Comment

“It is a pleasure to be able to review this project and to be given the opportunity to engage with you at this stage. My compliments on formulating a concise, cogent, and much-needed call for the study you are proposing. Studies of the scale and methodological rigor you present are much needed, and it is encouraging to see such competent researchers rise to the challenge. I was left wondering a couple of things regarding your study”

We thank Dr. Lobel for his largely positive take on our proposed study.

Point 1

The literature on violent video gaming has recently expanded into examining the influence of cooperative and competitive gaming. Given the proposed size of your study's sample, I think you are in the unique position to be able to add clarity and great value by including these as potential mediating variables. Was this a consideration of yours? And if so, what influenced your decision not to investigate these variables? (Related to this, your study already asks participants whether the game they have been playing was being played in single or multiplayer mode.)

This is a great question. We have checked and there is no ‘objective’ measure of prosocial content that we could code that would parallel the violent game content badging in the PEGI system. We believe this could be done, if a reliable subjective coding system could be crafted. In particular, because we will be collecting the prosocial behaviour subscale of the SDQ (2) we invite Dr. Lobel to investigate this alternative research question as a secondary use of the data the proposed study will generate.

Point 2

I commend your choice to rely on the PEGI ratings. I wonder whether you think it is important to distinguish between different types of violence ratings applied by PEGI (e.g. "cartoon violence" vs. "realistic violence").

In line with Dr. Etchells (point 2), and Dr Lobel’s suggestion here, we have included an additional sensitivity analysis that tests an additional, more granular, coding scheme for violent game content.

Point 3

“Based on past studies conducted by myself and my colleagues, you may want to ask for more specificity when asking participants to nominate the games they played most over the past month. While devoted gamers are likely to be very specific about the games they played, more casual gamers are apt to list generic titles (e.g. "Mario") or to simply list the name of websites that host large numbers of simple Flash games and mini-games. You may therefore wish to give your participants a bit more directive with this item.”

This is an excellent point. We have now extended our instructions for this item. New content is in italics. Please name the games you played most in the past month *and please try to be specific. For example, instead of typing “Mario Kart” or “COD” please type the specific game name, for example “Mario Kart 8” or “Call of Duty: Black Ops: Declassified”.*

Point 4

“I also miss the wording for how you will ask participants to describe whether they play single player or multiplayer (described as "variable 4" on page 5, but not listed in Appendix B).”

We apologise for this omission. It is now included in the materials in Appendix B.

Point 5

“Finally, some suggestions for very minor wording tweaks: -Page 1, Paragraph 1. "Following preregistered analysis plan multiple regression analyses will test..." Add a comma after "plan".”

We have made this change.

Point 6

“Page 4, Paragraph 3. "This survey-based study will... measure youth aggression with caregiver responses to a widely used in behavioural screening questionnaire..." I think this would read better with the word "questionnaire" moved to after "used".”

We have made this change.

Closing thoughts

“Thank you for your work, Best of luck, Adam Lobel, PhD”

We thank Dr. Lobel for raising these points with respect to our work and for signing his review.

Summary

In sum, we attempted to address the points you highlighted and those made by the reviewers. In nearly all cases we were able to directly address the issues with changes to the paper, study materials, and analytic strategy. We believe these have all been improved as a result. These reviewer inputs also led to conceptual tightening in our introduction. We were glad you and the reviewers were largely enthusiastic about this research, and hope you will find the paper adequately revised so that we can proceed with conducting our study.

Sincerely,

Dr. Andrew K. Przybylski
Oxford Internet Institute
University of Oxford
1 St Giles'
Oxford, UK, OX1 3JS

Dr. Netta Weinstein
School of Psychology
Cardiff University
70 Park Place
Cardiff, UK, CF10 3AT

References

1. Bryant FB, Smith BD. Refining the Architecture of Aggression: A Measurement Model for the Buss–Perry Aggression Questionnaire. *J Res Personal*. 2001 Jun;35(2):138–67.
2. Goodman R, Ford T, Simmons H, Gatward R, Meltzer H. Using the Strengths and Difficulties Questionnaire (SDQ) to screen for child psychiatric disorders in a community sample. *Br J Psychiatry J Ment Sci*. 2000 Dec;177:534–9.
3. Hilgard J, Engelhardt CR, Rouder JN. Overstated evidence for short-term effects of violent games on affect and behavior: A reanalysis of Anderson et al. (2010). *Psychol Bull*. 2017;143(7):757–74.
4. Przybylski AK, Ryan RM, Rigby CS. The Motivating Role of Violence in Video Games. *Pers Soc Psychol Bull*. 2009 Feb 1;35(2):243–59.
5. Westfall J, Yarkoni T. Statistically Controlling for Confounding Constructs Is Harder than You Think. Tran US, editor. *PLOS ONE*. 2016 Mar 31;11(3):e0152719.
6. Lakens D. TOST equivalence testing [Internet]. 2017 [cited 2017 Aug 10]. Available from: <https://osf.io/q253c/>
7. Kutner L, Olson CK. *Grand theft childhood: the surprising truth about violent video games and what parents can do*. 1st Simon & Schuster hardcover ed. New York: Simon & Schuster; 2008. 260 p.
8. Lenhart. *Teens, Social Media & Technology Overview 2015* [Internet]. Pew Research Center: Internet, Science & Tech. 2015 [cited 2016 Feb 8]. Available from: <http://www.pewinternet.org/2015/04/09/teens-social-media-technology-2015/>

Appendix B

October 26, 2018

Dear Alice Power and the editorial team,

Thank you very much for your 'in principle acceptance' of our stage one registered report originally titled: "Violent Video Game Engagement and Adolescents' Aggressive Behaviour: A Registered Report". I apologise for the delay in returning our stage two submission to RSOS. I (Dr. Przybylski) took on a new role in my department and it has been a larger drain on my time than I anticipated. I am sorry for the additional administrative burden this places on you and the team and I would be happy to recontact the named reviewers and directly request their assistance to get this process back on track.

We have completed the work and submitted the updated paper along with our data and analysis code (on the Open Science Framework: https://osf.io/d6wab/?view_only=bb7a969c2f0a420b8a0281b02aab5f12) and has been uploaded to RSOS as an ESM. The retitled the work: "Violent Video Game Engagement is not Associated with Adolescents' Aggressive Behaviour: Evidence from a Registered Report" has a changed title in line with the findings we derived after we conducted the study, as planned, in March of 2018.

Although our results are in the paper, we would like to note some of our findings briefly. We successfully recruited a sample of 1,004 adolescents and matching number of caregivers to participate. Our coding of game content using ratings went well and in terms of the general gaming literature we found reasonable patterns including gender-based game preference and engagement and our check question performed as expected. Findings diverged from our expectations as we considered our focal hypotheses concerned the effect of recent violent video game play on caregiver reports of aggressive behaviour. On the basis of the pre-existing literature (and stage one reviewer guidance) we set a minimum threshold for the relation, $r = .21$ (95% CI = .20 to .22) after adjusting our confirmatory tests for confounding variables. We found a substantially smaller effect than anticipated, $r = .01$ (95% CI = -.08 to .10). We observed this same pattern across a series of exploratory and

analyses including alternate predictor and outcome variables. We found no support for the core idea we were testing. From this we concluded the effect reported in meta-analytic studies might themselves be a byproduct of a range of flexible research methods original research regularly uses. In our discussion we aim to provide a context for understanding our pattern of findings with respect to the broader violent video game literature.

Sincerely,

Dr. Andrew K. Przybylski
Oxford Internet Institute
University of Oxford
1 St Giles'
Oxford, UK, OX1 3JS

Dr. Netta Weinstein
School of Psychology
Cardiff University
70 Park Place
Cardiff, UK, CF10 3AT

Appendix C

Dear Alice Power and the editorial team,

Thank you very much for your ‘in principle acceptance’ of our stage two registered report titled: “Violent Video Game Engagement is not Associated with Adolescents’ Aggressive Behaviour: Evidence from a Registered Report”. We have made the minor revisions suggested by the associate editor and reviewers. We address each in turn below and hope that you find the work fully revised and suitable for publication in RSOS.

Dr. Andrew K. Przybylski
Oxford Internet Institute
University of Oxford
1 St Giles’
Oxford, UK, OX1 3JS

Dr. Netta Weinstein
School of Psychology
Cardiff University
70 Park Place
Cardiff, UK, CF10 3AT

Associate Editor Comments to Author (Professor Chris Chambers)

General View

“The Stage 2 manuscript was returned to three of the expert reviewers who assessed the Stage 1 protocol. All are very positive about the submission and judge that the Stage 2 criteria are largely met. However, the reviewers offer some useful final suggestions to consider in the interpretation and conclusions (e.g. the point about considering $d=0.5$ as a meaningful cutoff) as well as some other minor corrections and clarifications. A minor revision is therefore recommended. Provided the revision addresses the reviewers’ points, final Stage 2 acceptance should be forthcoming without requiring further in-depth review.”

We are grateful for this feedback and have addressed these points and each reviewer comment in the current draft of the letter and paper.

Reviewer 1 (Professor Chris Ferguson)

General View

“The current study is the completion of a registered research report looking at violent game playing and aggression in youth. The study appears to have followed the registration, managed to obtain an impressive sample and has been conducted in a thorough and convincing manner. I am sure that this study will have a major impact on debates about violent game impacts. I have just a few very minor observations”

Thank you for noting this.

Point 1

“The authors mention the APA Task Force Report from 2015, but they might also want to note that the APA’s own media psychology division in 2017 released their own policy statement implicitly rebuking the APA as a whole and recommending against linking violent games to serious aggression in real life.”

Thank you for this important point. We have added this reference to the same section on p. 2 of the manuscript.

Point 2

“The authors do a nice job summarizing a complex research field as well as important methodological issues for this field. The only place they make an error is in concluding (as they repeat in the discussion) that there is only 1 previous preregistered study of VVG effects. The McCarthy study is certainly a worthwhile one, but it’s not the only preregistered study. I’ve done several myself dating back to 2015, and I believe James Ivory has at least one.”

We have now added references, as appropriate, to the two Ferguson papers and one Ivory paper we could find, and correctly claim this paper followed these earlier studies (see p. 4 and 14).

Point 3

“It’s good that the authors note some of the problems with methodological flexibility in prior research. The example of the Singapore dataset is a striking one.”

We agree, we were quite surprised by the ways by which this dataset has been segmented without attribution.

Point 4

“The statistical analyses are all well done. Their method of calculating VVG exposure is clearly cutting edge and they are right not to use participant reports.”

We thank the reviewer for noting this.

Point 5

“As a minor issue Page 10, p-values for ESB ratings may be swapped for the linear and parabolic outcomes.”

Great catch, thank you! Now swapped on p. 10.

Concluding Remarks

“Other than that, this is ready to go. I congratulate the authors on excellent work!”

We thank Professor Fergusson his expert observations and suggestions.

Reviewer 2

General View

“This Stage 2 manuscript is written cleanly, appeared to followed the agreed-upon methods and analyses, and has properly interpreted the results. I believe the authors have done an excellent job and have no issues with the manuscript being published as-is. I have three minor points that the authors could consider.”

Thank you for this.

Point 1

“First, as an additional positive control, I would note that Trait Physical Aggression was positively

associated with participants' aggressive behavior, $r = .62, p < .001$. This adds credence to the notion that participants were responding sensibly and that the measure of aggressive behavior is valid."

On p. 8 we now state "Importantly, trait-level aggression as reported by adolescents was strongly correlated with higher caregiver reports of aggressive behaviour as measured in the SDQ, $r = .62, p < .001$, supporting the notion that our criterion measure was a valid indicator of aggressive behaviour"

Point 2

"Second, it seems buried in the manuscript that aggressive behaviors are positively associated with violent game engagement ($r = .08$), overall game engagement ($r = .11$), and subjective game engagement ($r = .25$). I have no doubt that these numbers will be seized upon by proponents of the "violent video game-aggression effect" to argue that your conclusions are misguided."

This is a very interesting point that speaks to the larger challenges facing a literature largely based on post hoc analyses. If the work was not preregistered and we were so inclined, we might capitalise on chance and publish the work as in terms of legitimating just such a position. We have added a paragraph to the discussion considering how we should think about these correlations with respect to robust exploratory approaches such as specification curve analysis.

Point 3

"Finally, I am a bit ambivalent about the paragraph in arguing for adopting $d = +0.50$ as the benchmark for a meaningful effect. I can easily see effects being smaller than that being meaningful, especially if that effect is describing an aggressive behavior. I would consider just taking that paragraph out of the manuscript."

In response, we have removed this suggestion and we more generally argue the literature should work toward adopting effect sizes which are practically, not just statistically, significant (see p. 12). We do not offer what this number should be. To the reviewer's point and the broader idea we are currently conducting a series of large-scale simulation studies to arrive at data-driven workflow to move beyond hard and fast rules to ones that take intrinsic characteristics of the data into consideration.

We thank Reviewer 2 for his or her continued insight and attention to detail.

Reviewer 3

General View

"The authors present a well-powered study that has been conducted and analysed in accordance with the procedures initially outlined in their Stage 1 submission, and pass the outcome-neutral conditions. I note that a number of minor changes to the text of the introduction have been made, but these are restricted to synonyms and clarifications, and do not alter the substance of this section in any way. No changes were made to the stated hypotheses, and the rationale remains that same as that outlined at Stage 1. As far as I can ascertain, the authors clearly and strictly adhered to their registered experimental methodology and procedures. They were unable to calculate the inflection points for violent gaming effects, as the initial parabolic term was non-significant, making this sensitivity analysis redundant. This is fine, and is in line with their Stage 1 outline."

The authors conducted two additional exploratory analyses which were not registered in the Stage 1 submission; these are clearly denoted as such at Stage 2. Both analyses – that is, using US-based ratings (ESRB) as opposed to the European-based (PEGI) ratings used in the pre-registered analysis, and assessing prosocial behaviour (as opposed to aggressive behaviour) as an outcome measure – make sense within the context of the theoretical rationale, and add a further layer of robustness to

their analyses by (a) making the findings more internationally-relevant, and (b) assessing for positive vs. negative effects of video game play. In this regard, I am satisfied that the additional analyses are methodologically sound, and are clearly relevant.

The authors also provide a clear and objective overview of their results in the discussion section. The conclusions that they draw – that adolescent ‘violent’ video game play has no statistically-significant (or practically significant) effects on either aggressive or prosocial behaviour – is clearly borne out by the data that they have presented, and they present a well-reasoned and evidence-backed summary of the study. As I noted in my Stage 1 reviews, this study presents a much-needed addition to an area of research that, in general, would benefit hugely from further preregistered analyses (and RR submissions). Aside from the results themselves adding significant value to our understanding of the topic, the fact that the authors have demonstrated that it is clearly achievable to conduct a Registered Report concerning video games effects will hopefully provide an impetus for other researchers to hold themselves to similarly high standards in the future.”

We thank Reviewer 3 for his or her careful reading of our manuscript and for understanding what we think is a piece of research which substantively advances an important, if controversial, area of behavioural science. We are actively planning on improving and expanding on our approach.